# 6mer seed toxicity in tumor suppressive microRNAs

Quan Q. Gao[1], William E. Putzbach[1], Andrea E. Murmann[1], Siquan Chen[2], Aishe A. Sarshad[3], Johannes M. Peter[4], Elizabeth T. Bartom [5], Markus Hafner [3] & Marcus E. Peter [1,5]

Many small-interfering (si)RNAs are toxic to cancer cells through a 6mer seed sequence (positions 2–7 of the guide strand). Here we performed an siRNA screen with all 4096 6mer seeds revealing a preference for guanine in positions 1 and 2 and a high overall G or C content in the seed of the most toxic siRNAs for four tested human and mouse cell lines. Toxicity of these siRNAs stems from targeting survival genes with C-rich 3′UTRs. The master tumor suppressor miRNA miR-34a-5p is toxic through such a G-rich 6mer seed and is upregulated in cells subjected to genotoxic stress. An analysis of all mature miRNAs suggests that during evolution most miRNAs evolved to avoid guanine at the 5′ end of the 6mer seed sequence of the guide strand. In contrast, for certain tumor-suppressive miRNAs the guide strand contains a G-rich toxic 6mer seed, presumably to eliminate cancer cells.

[1] Department of Medicine, Division Hematology/Oncology, Northwestern University, Chicago, IL 60611, USA. [2] Cellular Screening Center, Institute for Genomics & Systems Biology, The University of Chicago, Chicago, IL 60637, USA. [3] Laboratory of Muscle Stem Cells and Gene Regulation, NIAMS, NIH, Bethesda, MD 20892, USA. [4] DigiPen Institute of Technology, Redmond, WA 98052, USA. [5] Department of Biochemistry and Molecular Genetics, Northwestern University, Chicago, IL 60611, USA. These authors contributed equally: Quan Q. Gao, William E. Putzbach. Correspondence and requests for materials should be addressed to M.E.P. (email: m-peter@northwestern.edu)

RNA interference (RNAi) is a form of post-transcriptional regulation exerted by 19–21 nt long double-stranded RNAs that negatively regulate gene expression at the mRNA level. RNAi-active guide RNAs can come from endogenous siRNAs and micro(mi)RNAs. For an miRNA, the RNAi pathway begins in the nucleus with transcription of a primary miRNA precursor (pri-miRNA)[1]. Pri-miRNAs are first processed by the Drosha/DGCR8 microprocessor complex into pre-miRNAs[2], which are then exported from the nucleus to the cytoplasm by Exportin-5[3]. Once in the cytoplasm, Dicer processes them further[4,5] and these mature dsRNA duplexes are then loaded into Argonaute (Ago) proteins to form the RNA-induced silencing complex (RISC)[6]. The sense/passenger strand is ejected/degraded, while the guide strand remains associated with the RISC[7]. Depending on the degree of complementarity between the guide strand and its target, the outcome of RNAi can either be target degradation—most often achieved by siRNAs with full complementarity to their target mRNA[8]—or miRNA-like cleavage-independent silencing, mediated by deadenylation/degradation or translational repression[9]. The latter mechanism can be initiated with as little as six nucleotide base-pairing between a guide RNA's so-called seed sequence (positions 2–7) and fully complementary seed matches in the target RNA[10,11]. This seed-based targeting most often occurs in the 3′UTR of a target mRNA[12,13].

A number of miRNAs function either as tumor suppressors or as oncogenes[14]. Their cancer-specific activities are usually explained by their identified targets, being oncogenes or tumor suppressors, respectively[14]. Examples of targets of tumor-suppressive miRNAs are the oncogenes Bcl-2 for miR-15/16[15] and c-Myc for miR-34a[16]. While many miRNAs have been reported to have both tumor suppressive and oncogenic activities depending on the cancer context, examples for widely established tumor-promoting miRNAs are miR-221/222, miR-21, miR-155, and members of the miR-17~92 cluster, or its paralogues miR-106b~25 and miR-106a~363[17,18]. In contrast, two of the major tumor-suppressive miRNA families are miR-15/16 and the p53 regulated miR-34a/c and miR-34b[19].

We recently discovered that many si- and shRNAs can kill all tested cancer cell lines through RNAi by targeting the 3′UTRs of critical survival genes (SGs)[20]. We called this mechanism DISE (for death induced by SG elimination). Cancer cells have difficulty in developing resistance to this mechanism both in vitro and when treated in vivo[21]. We reported that a 6mer seed sequence in the toxic siRNAs is sufficient for effective killing[20]. We have now performed a strand-specific siRNA screen with a library of individual siRNAs representing all 4096 possible 6mer seed sequences in a neutral RNA duplex. This screen, while based on siRNA biochemistry, was not designed to identify targets that are degraded through siRNA-mediated slicing activity but to identify toxicity caused by moderately targeting hundreds of genes required for cell survival in a mechanism similar to miRNA-induced silencing.

We report that the most toxic 6mer seeds are G-rich with a G enrichment towards the 5′ end targeting SGs with a high C content in their 3′UTR in a miRNA-like manner. Many tumor-suppressive miRNAs such as miR-34a-5p but none of the established oncogenic miRNAs contain G-rich 6mer seeds and most of miR-34a-5p's toxicity comes from its 6mer seed sequence. Mature miRNAs from older and more conserved miRNAs contain less toxic seeds. We demonstrate that for most miRNAs the more abundant mature form corresponds to the arm that contains the less toxic seed. In contrast, for major tumor-suppressive miRNAs, the mature miRNA is derived from the arm that harbors the more toxic seed. Our data allow us to conclude that while most miRNAs have evolved to avoid targeting survival and housekeeping genes, certain tumor-suppressive miRNAs function to kill cancer cells through a toxic G-rich 6mer seed targeting the 3′UTR of SGs.

## Results

**Identifying the most toxic 6mer seeds.** To test whether certain 6mer seeds present in the guide strand of an siRNA affect cancer cell survival, we recently designed a neutral 19mer oligonucleotide scaffold with two nucleotide 3′ overhangs, and we demonstrated that modifying an siRNA strand at positions 1 and 2 by 2′-O-methylation (OMe) completely blocks its loading into the RISC[22]. Different 6mer sequences can be inserted at positions 2–7 of the guide strand with the designated passenger strand modified by OMe (two red Xs in Fig. 1a). Transfection efficiency and conditions were optimized for each cell line used. To determine the general rules of seed-based toxicity, we individually transfected 4096 siRNAs with all possible 6mer seed sequences in this 19mer scaffold into two human, HeyA8 (ovarian cancer) and H460 (lung cancer), and two mouse cell lines M565 (liver cancer) and 3LL (lung cancer). This allowed us to rank all 4096 6mer seeds according to their toxicity (Fig. 1b, Supplementary Data 1, and 6merdb.org). The congruence between the results of the two human cell lines and the two human and two mouse cell lines was quite high ($r = 0.68$ and $0.73$, respectively; Fig. 1c), suggesting that many siRNAs were toxic through a mechanism independent of cancer origin and species. Toxicity was caused by the different 6mer seeds in the guide strand. An siRNA duplex highly toxic to all cell lines (#2733, HeyA8 cell viability = 1.4%) strongly inhibited cell growth and reduced cell viability of HeyA8 cells only when the passenger strand but not when the guide strand was modified by the OMe modification. Toxicity was completely blocked when the guide strand was modified (Supplementary Fig. 1a). The toxicity was due to RNAi as knockdown of AGO2 abolished the toxicity of two of the most toxic siRNAs (Supplementary Fig. 1b).

We previously reported that the CD95 ligand (CD95L) coding region (CDS) is enriched in sequences that when converted into si- or shRNAs are toxic to cancer cells[20] and most recently that the CD95L mRNA itself is toxic to cells[23]. We now report a substantial correlation between the most toxic CD95L-derived shRNAs and the toxicity of their predicted 6mer seed (Fig. 1d), suggesting the CD95L-derived si/shRNAs kill cancer cells through 6mer seed toxicity. Consistent with this assumption we found that the 6mer seeds of four previously tested siRNAs derived from CD95L[24] in this screen were about as toxic as the full-length siRNAs, with siL3$^{Seed}$ being the most toxic followed by siL2$^{Seed}$ and no toxicity associated with siL4$^{Seed}$ and siL1$^{Seed}$ (Fig. 1b, c). Our recent analysis suggested that the toxic si/shRNAs act like artificial miRNAs by targeting the 3′UTR of mRNAs[20].

**6mer seeds enriched in G at the 5′end are most toxic.** We noticed that the 6mer seeds in siL3 and siL2 guide strands have a higher G content than the ones in siL4 and siL1 (Fig. 1b). By analyzing the screen results of all four cell lines (Supplementary Fig. 2), we found that a high G content in the seed correlated better with toxicity than a high C content. Almost no toxicity was found with seeds with a high A content. To test the effect of nucleotide content on toxicity directly, we retested the 19 seed duplexes with the highest content (>80%) for each of the four nucleotides in the four cell lines (Fig. 2). The reanalysis also allowed us to determine the reproducibility of the results obtained in the large screens (which for technical reasons had to be performed in three sets). All data on the four cell lines were highly reproducible especially for the most toxic seeds (Supplementary Fig. 3a). When the data on the four cell lines were compared, it became apparent that in all cell lines, the G-rich seeds were by far the most toxic followed by the C-rich, U-rich, and A-rich seeds (Fig. 2a). This indicates it is mostly the G content that determines toxicity.

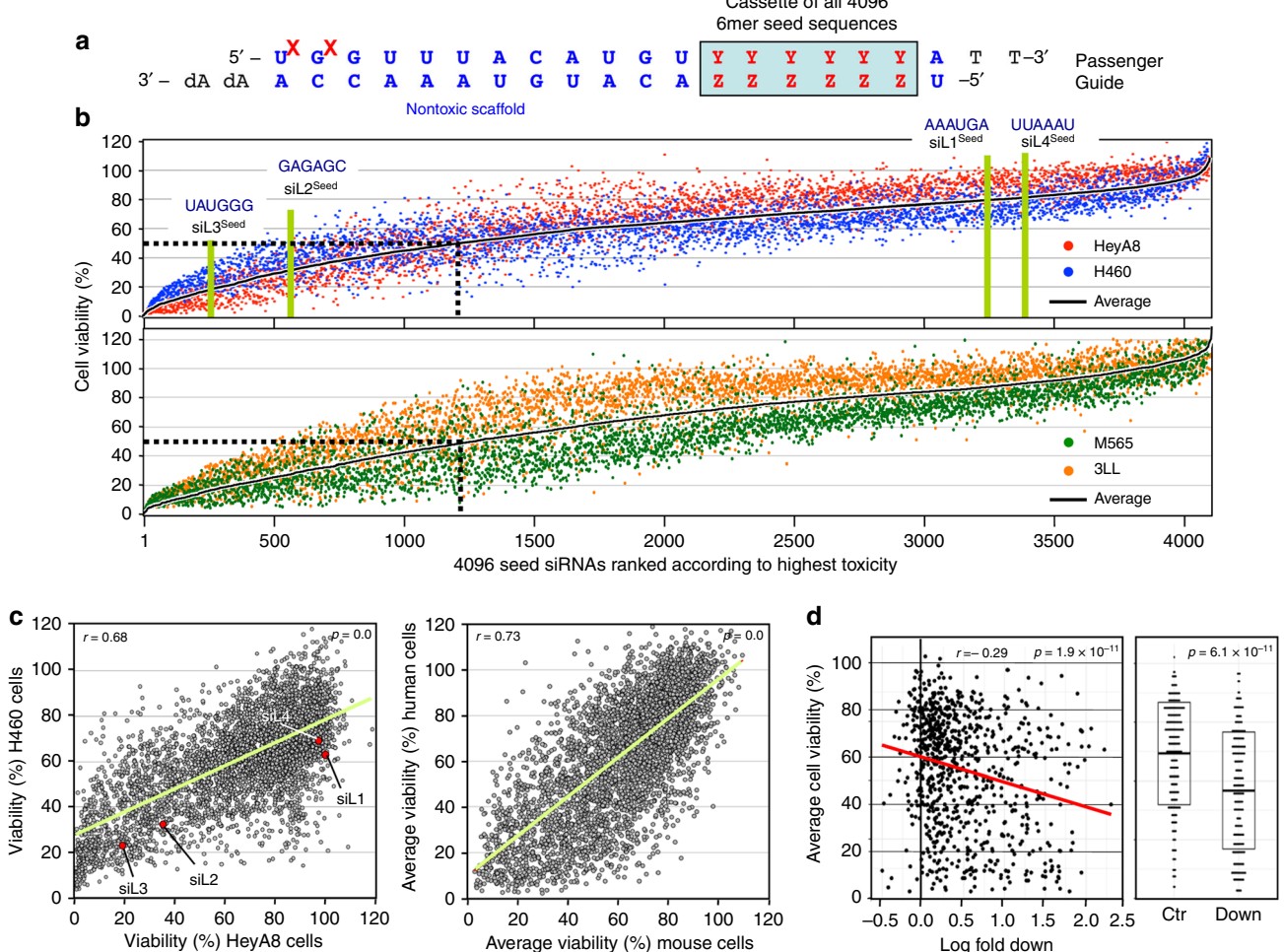

**Fig. 1** A comprehensive screen identifies the most toxic 6mer seeds. **a** Schematic of the siRNA backbone used in the 4096 seed duplexes toxicity screen. Red X: 2′-O-methylation modification; blue letters: constant nucleotides; red letters: variable 6mer seed sequence. **b** Results of the 4096 6mer seed duplex screen in two human (top) and two mouse (bottom) cell lines. Cells were reverse transfected in triplicates in 384-well plates with 10 nM of individual siRNAs. The cell viability of each 6mer seed duplex was determined by quantifying cellular ATP content 96 h after transfection. All 4096 6mer seeds are ranked by the average effect on cell viability of the four cell lines from the most toxic (left) to the least toxic (right). Rankings of the 6mer seeds of four previously characterized CD95L-derived siRNAs (siL1, siL2, siL3, and siL4) are highlighted in green. We consider an siRNA highly toxic if it reduces cell viability 90% or more and moderately toxic if it reduces cell viability 50% or more (black stippled line). **c** Regression analysis showing correlation between the 6mer seed toxicity observed in the human lung cancer cell line H460 (y-axis) and the matching 6mer toxicity observed in the human ovarian cancer cell line HeyA8 (x-axis) (left) and average toxicity in the two human cell lines (y-axis) and two mouse cell lines (x-axis) (right). p-Values were calculated using Pearson correlation analysis. **d** Left: Correlation between the $\log_{10}$ (fold down underrepresentation) of all possible shRNAs that can be derived from the mRNA CDS sequence of CD95L following their expression from a DOX-inducible lentiviral vector[20] and the toxicity of their seed sequences as determined in a 4096 arrayed siRNA screen (average of both human cell lines). Right: Difference in average seed toxicity between the 137 CD95L-derived shRNAs downregulated at least five-fold (=toxic) in this screen compared to a size-matched group of 137 shRNAs that were the least altered in abundance in that screen. Pearson correlation coefficient is given as well as p-value (left) and p-value in analysis on the right was calculated using unpaired two-sided t-test. The center crossbar of the box plot represents the mean, and the upper/lower boundaries demark ±1 standard deviation

Most genome-wide siRNA libraries designed to study functions of individual genes are highly underrepresented in G and C to increase RNAi specificity[25] (Supplementary Fig. 3b, left panel). In contrast, our complete set of 6mer seed duplexes exhibits no nucleotide composition bias, allowing us to test the contributions of all four nucleotides in each of the six seed positions (Supplementary Fig. 3b, right panel). To determine the nucleotide content of the most toxic seeds, we determined the frequency of each nucleotide at each of the six positions of either the 200 most or 200 least toxic seed duplexes for each of the two human and the two mouse cells (Fig. 2b, Supplementary Data 1). We found that a high G content towards the 5′ end of the seed and a C in position 6 was most toxic (Fig. 2b and Supplementary Fig. 3c). In contrast, non-toxic seeds were A- and U-rich especially when

positioned at the 5′ end. The rules of toxicity that emerged are almost identical between human and mouse, suggesting evolutionary conservation. This can also be explored at 6merdb.org.

**Toxic siRNAs target C-rich housekeeping genes**. We previously showed that si- and shRNAs are toxic through 6mer seed toxicity preferentially targeting hundreds of genes critical for cell survival[20]. We had developed a Toxicity Index (TI), a simple tool to predict the most toxic seeds based on the ratio of putative seed match occurrences in the 3′UTR of set of SGs versus a set of genes not required for cell survival (non-SGs). We now compared the TI with our experimentally determined 6mer seed toxicity in the four cell lines screened (Supplementary Fig. 4) and found a

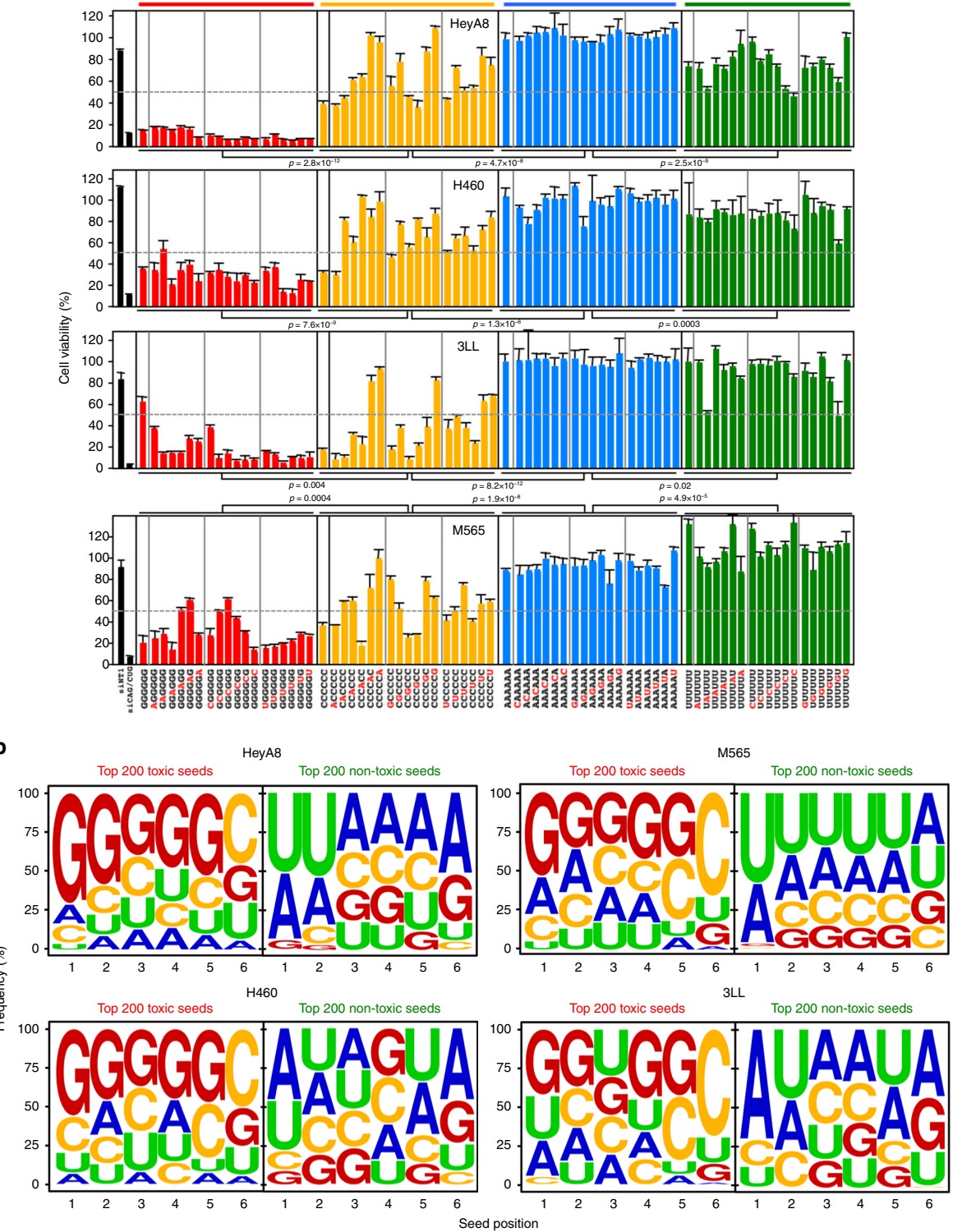

**Fig. 2** The most toxic seeds are G rich. **a** Cell viability of the 19 seed duplexes with the highest content (>80%) in the 6mer seed region for each nucleotide in two human and two mouse cell lines. Samples were analyzed in triplicate and mean ± SD for each sample is shown. *p*-Values between groups of duplexes were calculated using Student's *t*-test. siRNAs are considered to be toxic when viability is inhibited >50% (gray stippled line). **b** Nucleotide composition at each of the six seed positions in the top 200 most toxic (left) or the top 200 least toxic (right) seed duplexes in the four cell lines.

significant correlation between these two types of analyses, further supporting the mechanism of toxicity. Knowing that seed sequences rich in G are most toxic suggested the targeted genes carry C-rich seed matches in their 3′UTR. To be most stringent, we used a list of the 20 6mer seeds that were most toxic to both

HeyA8 and H460 cells (Supplementary Data 2). The G richness towards the 5′ end of the 6mers in these toxic seeds and the 5′ A/U richness of the non-toxic seeds was even more pronounced than in the top/bottom 200 most toxic seeds (Fig. 3a). We scored for the occurrence of seed matches to the 20 seeds

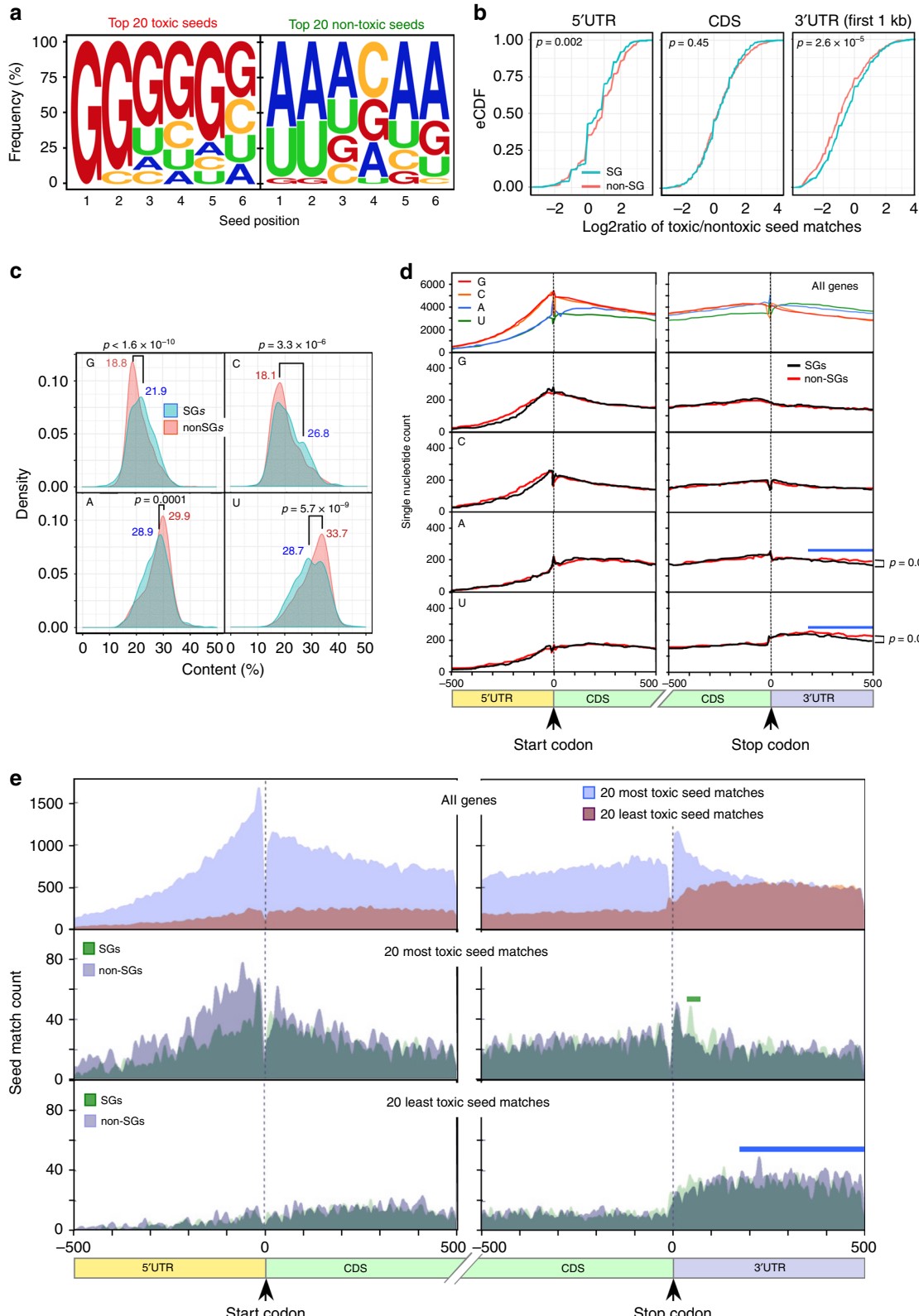

in each group in the 3′UTR, the CDS and the 5′UTR of a set of 938 critical SGs similar to one recently described[20] and an expression-matched set of 938 non-SGs. We found a significantly higher count ratio of toxic versus non-toxic seed matches in the 3′UTR of SGs when compared to non-SGs (Fig. 3b, right panel). Consistent with an miRNA-like function no such enrichment was

found when the CDS was analyzed (Fig. 3b, center panel). An inverse ratio of sequences complementary to the 6mer seeds of unknown significance was found in the 5′UTR (Fig. 3b, left panel). This result was very similar when we scored for seed matches to the 100 seeds most toxic and least toxic to both human cell lines (Supplementary Fig. 5a). An enrichment of the

**Fig. 3** Toxic G-rich seed-containing duplexes target housekeeping genes enriched in Cs. **a** Nucleotide composition of 20 seeds that are most and least toxic in both human cell lines (see Fig. 1b). **b** eCDF comparing the ratio of occurrences of the 20 most and least toxic 6mer seed matches in the mRNA elements of two sets of expression-matched survival genes and non-survival genes. Significance was calculated using a two-sample two-sided Kolmogorov–Smirnov (K–S) test. **c** Probability density plots comparing the nucleotide content between the groups of expression-matched SGs and non-SGs. *p*-Values were calculated using a two-sample two-sided K–S test comparing the density distribution of SGs and non-SGs. Relevant peak maxima are given. **d** Single-nucleotide frequency distribution in human mRNAs around the boundaries of the 5′UTR and the beginning of the CDS and the end of the CDS and the beginning of the 3′UTR (shown are 500 bases in each direction). Data are shown for either all human coding genes (top), or a set of 938 SGs or an expression-matched set of 938 non-SGs (bottom four panels). Blue horizontal bars, area of reduced A/U content in SGs. *p*-Values were calculated using a two-sample two-sided K–S test. **e** Distribution of the seed matches to the 20 most and least toxic 6mer seeds to human cells in human mRNAs around the boundaries of the 5′UTR and the beginning of the CDS and the end of the CDS and the beginning of the 3′UTR (shown are 500 bases in each direction). Data are shown for either all genes (top) or the expression-matched SGs and non-SGs (center and bottom). Green horizontal bar, area of enriched toxic seed matches in SGs compared to non-SGs. Blue horizontal bar, area of fewer toxic seed matches in SGs

exact seed matches in 3′UTRs was consistent with the overall higher C content of 3′UTRs of SGs when compared to non-SGs (different peak maxima in Fig. 3c). A metaplot analysis of the 500 bases upstream and downstream of the translational start and stop site of all human genes showed that as expected the 3′UTR was enriched in A and U (Fig. 3d, top). Interestingly, SGs had a lower A/U content in a region ~150–500 bases into the 3′UTR than expression-matched non-SGs (Fig. 3d, blue horizontal bar, bottom two panels). To determine where SGs are being targeted by toxic seed-containing siRNAs we again performed a metaplot analysis—this time plotting the locations of seed matches to the 20 6mer seeds that were most and least toxic to both human cell lines (Fig. 3e, an analysis with the 100 most/least toxic seeds is provided in Supplementary Fig. 5b). When analyzing all human coding genes we found the reverse complements of the most toxic seeds to be highly enriched at the beginning of the 3′UTR whereas the reverse complements of the least toxic seeds were underrepresented in this region (Fig. 3e, top). This effect was not due to a much higher G/C or lower A/U content in this region (Fig. 3d, top). A comparison of the location of these seed matches in the SGs and in expression-matched non-SGs confirmed this general trend; however, two differences between SGs and non-SGs became apparent: (1) non-SGs have more non-toxic seed matches ~150–500 bases into the 3′UTR (Fig. 3e, bottom, blue horizontal bar) maybe due to the higher A/U content of this region (Fig. 3d, two bottom panels). (2) SGs contain a small stretch at positions 42–65 into the 3′UTR (Fig. 3e and Supplementary Fig. 5b, center, green horizontal bar) that is enriched in seed matches for the most toxic seeds. This region in SGs seems to be a preferential target site for siRNAs carrying toxic G-rich seed sequences.

**miR-34a-5p kills cancer cells through its toxic 6mer seed**. The toxic siRNAs kill cancer cells through 6mer seed toxicity by a mechanism reminiscent of the function of miRNAs. To test whether actual miRNAs could kill cancer cells with the help of toxic 6mer seeds, we analyzed the seed toxicity determined in our screen for all known ~2600 mature miRNAs expressed as either the 3p or 5p arm (6merdb.org). While none of the 6mer seeds present in the predominant arm (guide strand) of the most oncogenic miRNAs (miR-221/222, miR-21, miR-155, the miR-17~92, miR-106b~25, and miR-106a~36 clusters) were toxic (reduced viability >50%, stippled line in Fig. 4a), two of the major tumor-suppressive miRNA families, miR-15/16 and p53, regulated miR-34a/c and miR-34b contained toxic seeds in the guide strand (Fig. 4a). This suggested these two families were killing cancer cells through toxic 6mer seeds. Interestingly, two other major tumor-suppressive families, let-7 and miR-200, did not contain toxic G-rich seeds in their guide strand, suggesting they may be tumor suppressive through other mechanisms, such as inducing and maintaining cell differentiation[26].

When transfecting the pre-miRs of miR-34a-5p, miR-15a-5p, and let-7a-5p into HeyA8 cells, the potency of these three miRNAs to reduce cell growth mimicked the toxicity of their 6mer seed-containing siRNAs (Fig. 4a, b). This suggested that a large part of their toxicity comes from the composition of the seed position 2–7. The most toxic seed in a major tumor-suppressive miRNA was present in miR-34a-5p/34c-5p, a master regulator of tumor suppression[27]. We directly compared the toxicity of pre-miR-34a-5p and its toxic seed in the neutral scaffold with blocked passenger strand (si34a-5p$^{Seed}$) in the same assays (Fig. 4c). Strikingly, the toxicity evoked by these two RNA species (assessed by growth inhibition and DNA fragmentation) was similar. Cells showed the typical morphology we found in cells dying from toxic siRNAs (Fig. 4d)[20,24,28]. To determine the contribution of the 6mer seed sequence of miR-34a-5p to its toxicity and the mode of cell death, we performed a RNA-Seq analysis on HeyA8 cells transfected with either miR-34a-5p or si34a-5p$^{Seed}$ (Fig. 5a, top) (Supplementary Data 3). The vast majority of genes were significantly up- and downregulated by both RNA duplexes (Fig. 5a, bottom). While miR-34a-5p targeted a subset of genes not affected by miR-34a-5p$^{Seed}$, the majority of differentially expressed genes (>78%) were downregulated >1.5-fold by both the premiR and the 6mer seed duplex (Fig. 5b, left). A Sylamer analysis is a unbiased approach allowing to identify which seed matches are enriched in the 3′UTRs of downregulated genes from RNA-seq data[29]. In this analysis both duplexes caused similar and highly effective downregulation of the mRNAs that carry a 6mer seed match (Fig. 5c). When the Sylamer analysis was performed with either 7mer or 8mer seeds, enrichment of seed matches was much less significant (Supplementary Fig. 6a), suggesting that most of the targeting by both RNAs only required a 6mer seed.

Consistent with this activity, targeting by both RNA duplexes resulted in a very similar reduction of SGs (Fig. 5d). The genes downregulated by both the premiR and the 6mer seed construct were highly enriched in genes involved in regulation of cell cycle, cell division, DNA repair, and nucleosome assembly (Fig. 5b, right). These GO terms were very similar to the ones we found enriched in downregulated genes in cells dying after transfection with CD95R/L-derived si/shRNAs containing toxic 6mer seeds[20]. In contrast, no such GO terms were found enriched when the same analysis was performed with the upregulated genes as control (Supplementary Fig. 6b). While both miR-34a-5p and si34a-5p$^{Seed}$ caused the most significant downregulation of genes carrying 8mers in their 3′UTR (Supplementary Fig. 6c), only the most highly downregulated genes that carry the shared 6mer seed match were grouped in a number of GO terms that are consistent with 6mer seed toxicity as previously reported[20] and barely any GO terms were shared among genes that contained 7 or 8mer seed matches (Supplementary Fig. 6d). All these data suggest that miR-34a-5p kills cancer cells using its toxic 6mer

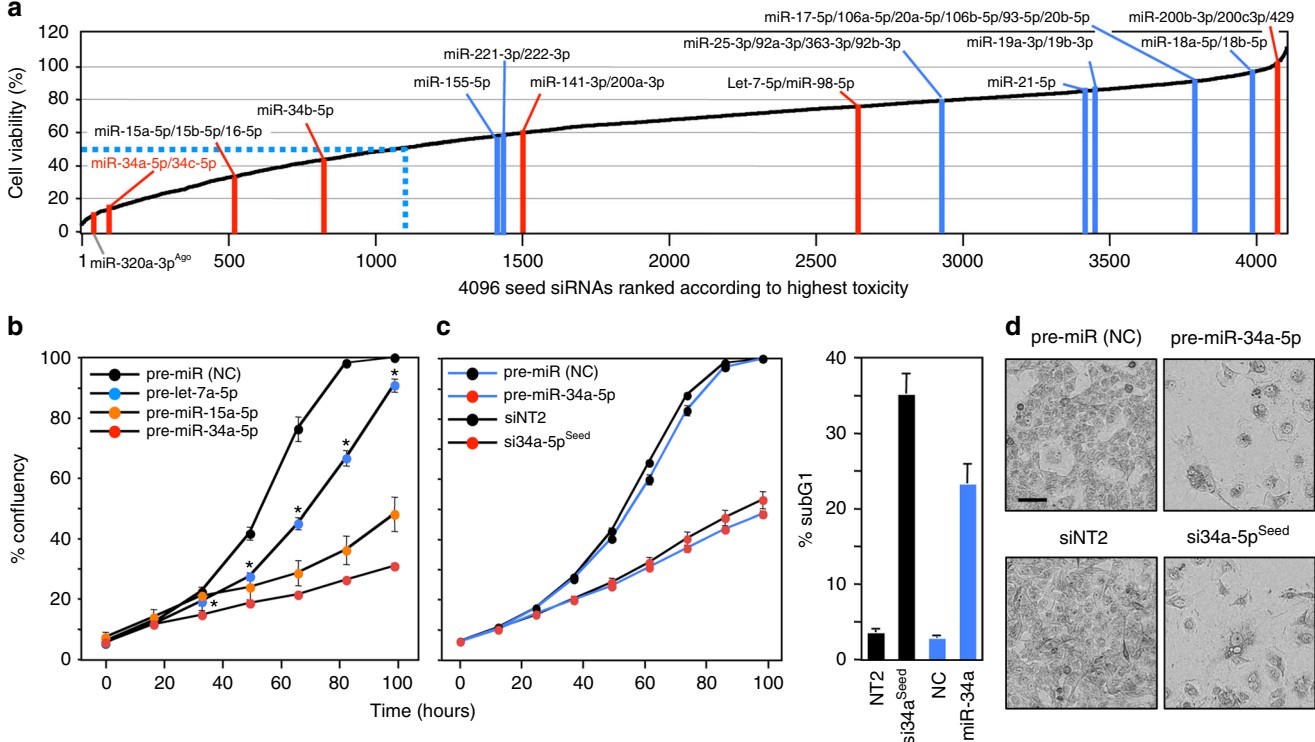

**Fig. 4** Tumor-suppressive miRNAs inhibit cancer cell growth via toxic 6mer seeds. **a** All 4096 6mer seeds ranked from the lowest average viability (highest toxicity) to the highest viability (lowest toxicity) according to the average of HeyA8 and H460 cells. Locations of 6mer seeds present in major tumor-suppressive (red) or tumor-promoting (blue) miRNAs are highlighted as individual bars. miRNAs are considered to be toxic when viability is inhibited >50% (blue stippled line). **b** Percent cell confluence over time of HeyA8 cells transfected with 5 nM of either the indicated tumor-suppressive miRNA precursors or an miRNA precursor non-targeting control. Data are representative of two independent experiments. Each data point represents mean ± SE of four replicates. *Two-way ANOVA p-value between cells treated with pre-miR-(NC) and pre-let-7a-5p is 0.0000. **c** Left: Percent cell confluence over time of HeyA8 parental cells transfected with either pre-miR-34a-5p or si34a-5p$^{Seed}$ and compared to their respective controls (pre-miR (NC) for pre-miR-34a-5p and siNT2 for si34a-5p$^{Seed}$) at 10 nM. Data are representative of two independent experiments. Each data point represents mean ± SE of four replicates. Right: % cell death of the same cells harvested 4 days after transfection. Data are representative of two experiments. Each data point represents mean ± SD of three replicates. **d** Morphology of HeyA8 cells transfected with 10 nM of either pre-miR-34a-5p or si34a-5p$^{Seed}$ compared to their respective controls 3 days after transfection

seed. While optimal miRNA targeting requires at least a 7mer seed interaction and also involves nucleotides at positions 13–16 of the miRNA[30], the cell death inducing activity of this tumor-suppressive miRNA may only require the 6mer seed.

**Toxic 6mer seed toxicity is shaping the miRNA repertoire.** Toxic 6mer seeds may be a driving force in miRNA evolution, whereby toxic seed sequences are either selected against—because they contribute to cell toxicity—or are preserved to operate as tumor suppressors. Based on the composition of toxic 6mer seeds and the enrichment of corresponding seed matches in SGs, we could now ask whether and when miRNAs that contain toxic G-rich sequences in positions 2–7 of their seeds evolved. When comparing all mature miRNA arms annotated in TargetScan Human 7, we noticed that miRNAs in highly conserved miRNA seed families contained 6mer seed sequences that were much less toxic in our screen than seeds in poorly conserved miRNAs (Fig. 6a, left panel and Supplementary Fig. 7a). Weakly conserved miRNA seed families would be expected to be younger in evolutionary age than highly conserved ones. Consistent with this assumption we found that the 6mer seeds of younger miRNAs (<10 million years old) were more likely to be toxic to cells than the ones of older miRNAs (>800 million years old)[31] (Fig. 6a, right panel and Supplementary Fig. 7b). Most importantly, when comparing miRNAs of different ages, it became

apparent that seeds of miRNAs over the last 800 million years were gradually depleted of G beginning at the 5′ end and eventually also affecting positions 3–5 until the oldest ones, where A and U had replaced G as the most abundant nucleotide in all six positions (Fig. 6b). These analyses indicated that most highly conserved miRNAs avoid G in potentially toxic seed positions. Interestingly, the most toxic seed sequences were found in miRtrons (Fig. 6c, Supplementary Fig. 7c, and 6merdb.org), miRNAs that are derived by splicing short introns[32]. Across all mature miRtrons we found G to be the most abundant nucleotide in position 2–7 (Supplementary Fig. 7d) and this region was also near the region in all miRtrons predicted to contain the 6mer sequences with the highest toxicity (Supplementary Fig. 7e).

miRNAs are expressed as pre-miRs and usually only one major species of mature miRNA (either the 5p or the 3p arm) is significantly expressed in cells produced from one of the two strands of the premiR stem[33]. Consistent with the assumption that cells cannot tolerate toxic 6mer seeds, we now examined across 780 miRNAs which have been shown to give rise to both a 3p and a 5p arm whether the more highly expressed arm contains a seed of lower toxicity than the lesser-expressed arm (Fig. 6d). We ranked the miRNAs according to the ratio of the 6mer seed toxicity associated with the guide arm to the lesser-expressed arm (Supplementary Data 4). When we labeled the major tumor-suppressive and oncogenic miRNAs, we noticed the highly expressed arm of most of the oncogenic miRNAs contained a

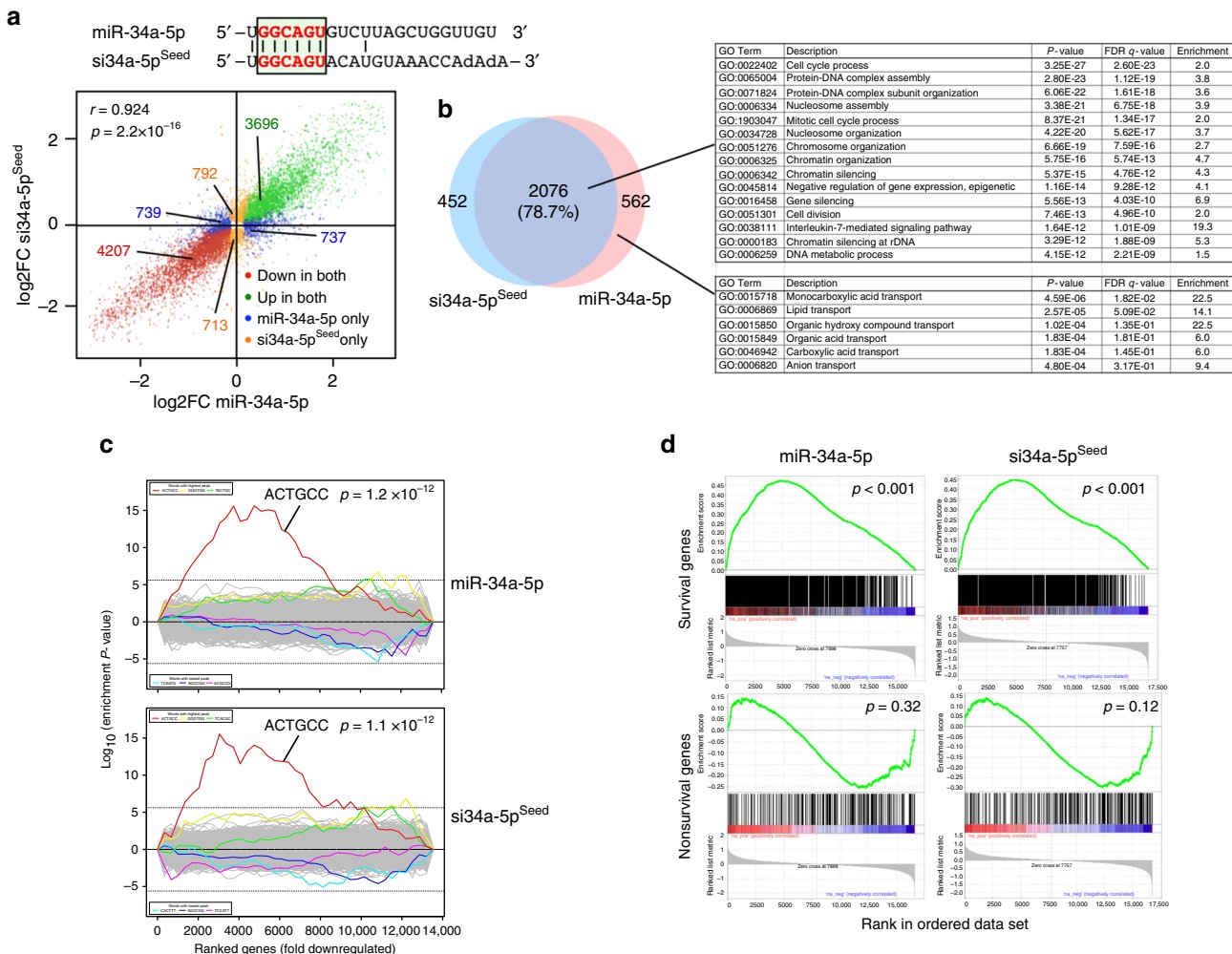

**Fig. 5** miR-34a-5p kills cancer cells through its toxic 6mer seed. **a** Top: Alignment of the sequences of miR-34a-5p and si34a-5p$^{Seed}$ with the 6mer highlighted. Bottom: Comparison of deregulated mRNAs (adjusted $p < 0.05$, RPM > 1) in HeyA8 cells 48 h after transfection with either miR-34a-5p or si34a-5p$^{Seed}$. Pearson correlation $p$-value is given. **b** Overlap of RNAs detected by RNA-Seq downregulated in HeyA8 cells (>1.5-fold) 48 h after transfection with either si34a-5p$^{Seed}$ or miR-34a-5p when compared to either siNT2 or a non-targeting pre-miR, respectively. Right: Results of a GOrilla gene ontology analyses of the genes downregulated in both cells transfected with miR-34a-5p or si34a-5p$^{Seed}$ (top, significance of enrichment <10$^{-11}$), or only in cells transfected with miR-34a-5p (bottom, significance of enrichment <10$^{-4}$). **c** Sylamer plots for the list of 3'UTRs of mRNAs in cells treated with either miR-34a-5p (top) or si34a-5p$^{Seed}$ (bottom) sorted from downregulated to upregulated. The most highly enriched sequence is shown which, in each case, is the 6mer seed match of the introduced 6mer seed. Bonferroni-adjusted $p$-values are shown. **d** Gene set enrichment analysis for a group of 1846 survival genes (top four panels) and 416 non-survival genes (bottom two panels)[20] after transfecting HeyA8 cells with either miR-34a-5p or si34a-5p$^{Seed}$. siNT1 and a non-targeting premiR served as controls, respectively. $p$-values indicate the significance of enrichment

6mer seed that was not toxic in our screen (Fig. 6d, blue dots). In contrast, for most of the tumor-suppressive miRNAs, the dominant arm contained a seed much more toxic than the lesser arm (Fig. 6d, red dots). The overall difference in ratio between the two groups of miRNAs was highly significant. A more detailed analysis of these data revealed that the three oncogenic miRNAs with the highest ratio in toxicity between their arms, miR-363, miR-92a-2, and miR-25, were almost exclusively expressed as the non-toxic 3p form (Fig. 6e, top). In contrast, the dominant arm of the three tumor-suppressive miRNAs, miR-34a, miR-34c, and miR-449b, contained the most toxic seed sequence (Fig. 6e, bottom). Interestingly, miR-449b has the same seed sequence as miR-34a and has been suggested to act as a backup miRNA for miR-34a[34]. These data are consistent with most tumor-suppressive miRNAs using 6mer seed toxicity to kill cancer cells and suggest that this mechanism developed over hundreds of millions of years.

**Genotoxic drugs upregulate toxic 6mer seed-containing miR-NAs.** Our data showing that miR-34a-5p contains a toxic 6mer seed, along with miR-34a being upregulated after genotoxic stress[19], led us to wonder whether miR-34a-5p would contribute to cell death induced by genotoxic drugs and whether this type of cell death shared similarities to the death observed in cells dying from toxic 6mer seed-containing si/shRNAs. This would be consistent with the observation that many genotoxic drugs induce multiple cell death pathways[35–39]. To compare cell death induced by different genotoxic agents with that of toxic si/shRNAs, we treated the p53 wild-type ovarian cancer cell line HeyA8 with doxorubicin (Doxo), carboplatin (Carbo), or etoposide (Eto) and performed a RNA-Seq analysis. Drug concentrations were chosen so that after 80 h, treatment would slow down cell growth and induce signs of stress without major cell death occurring to capture changes that could be causing cell death rather than being the result of it (Supplementary Fig. 8a). The morphological

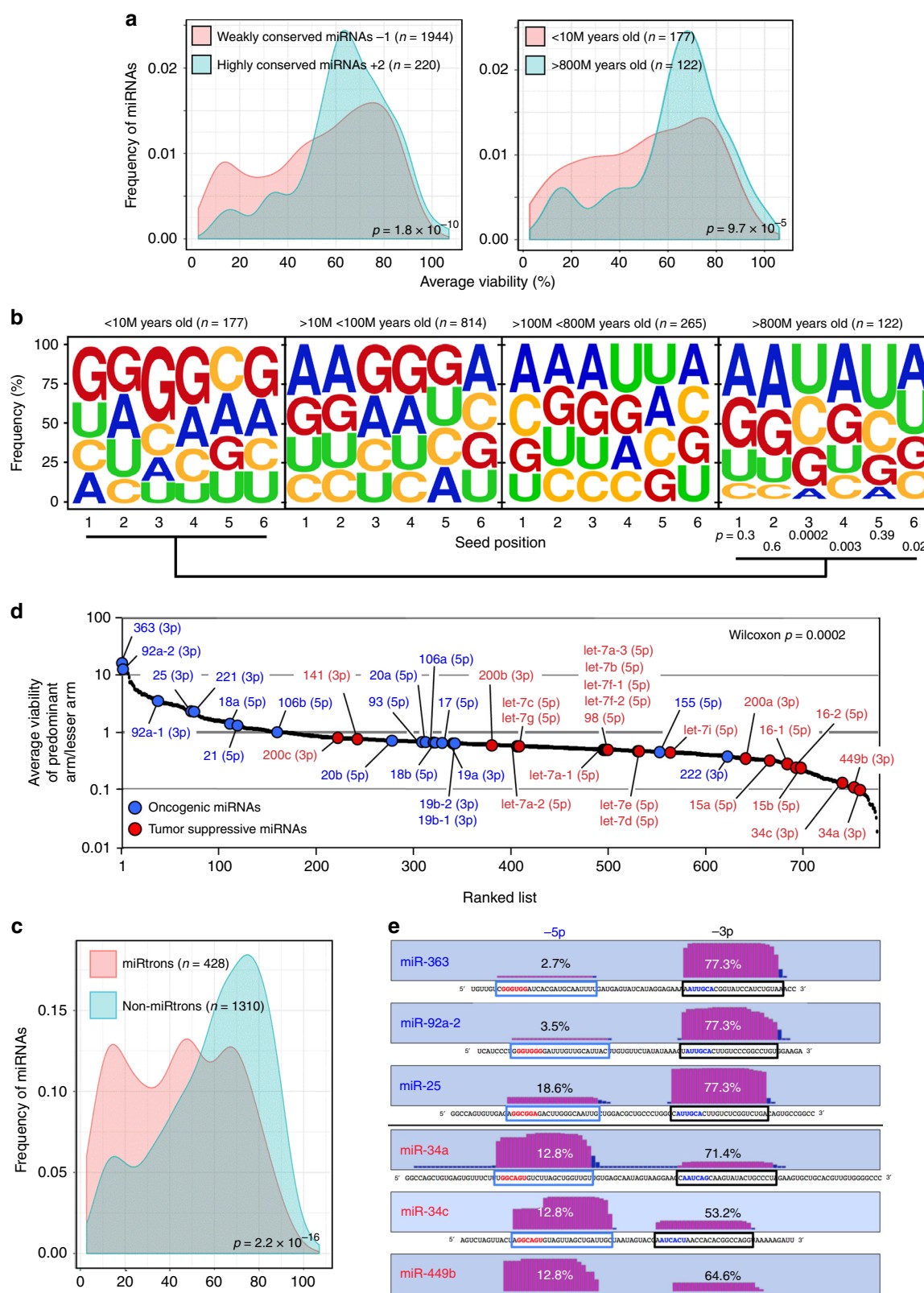

changes in the cells treated with the drugs were very similar to the ones seen in cells treated with si/shRNAs (Supplementary Fig. 8b), and similar to reported morphologies of cells treated with genotoxic drugs[40,41].

The ranked lists of downregulated RNAs isolated from HeyA8 cells treated with the three drugs were subjected to a gene set enrichment analysis (GSEA) to determine whether SGs were enriched in the downregulated genes (Supplementary Fig. 9a). There was strong enrichment of downregulated SGs towards the top of the ranked list. One hundred and two of the SGs were downregulated in cells treated with any of the three drugs (Supplementary Fig. 9b). In a DAVID gene ontology analysis,

**Fig. 6** Toxic 6mer seeds and the evolution of cancer regulating miRNAs. **a** Probability density plot of cell viability of the 6mer seeds of either highly conserved (from humans to zebrafish) or poorly conserved miRNA seed families (left panel, total number of mature miRNAs = 2164) or of very old (>800 (M) million years) miRNAs or very young (<10 million years) miRNAs (right panel, total number of miRNAs = 299). For the analysis on the right, miRNA arms with identical sequences (gene duplications) were collapsed and counted as one arm. Two-sample two-sided K–S test was used to calculate p-values. **b** Change in nucleotide composition in the 6mer seeds of miRNAs of different ages. Significance of change in nucleotide composition at each of the six seed positions between the youngest and oldest miRNAs was calculated using a Fisher's exact test. Note: the oldest miRNAs also contain tumor-suppressive miRNAs with high G content in positions 1 and 2, which may be the reason the analysis in these two positions did not reach statistical significance. **c** Probability density plot of cell viability of the 6mer seeds of mature miRtrons or non-miRtrons. miRNAs with identical sequences (gene duplications) were collapsed and counted as one seed. Two-sample two-sided K–S test was used to calculate p-value. **d** Seven hundred and eighty miRNAs (Supplementary Data 4) ranked according to the ratio of viability of the seed (as determined in the seed screen) of the guide strand and the lesser-expressed arm. Established oncogenic miRNAs are shown in blue; tumor-suppressive miRNAs are shown in red. The guide strand is given for each miRNA (in parenthesis). p-Value of the distribution of oncogenic versus tumor-suppressive miRNAs was calculated using Wilcoxon rank test. **e** Cumulative read numbers from the 5p or the 3p arm (according to miRBase.org) of three oncogenic and three tumor-suppressive miRNAs with the highest (top three) or a very low ratio of the viability of the guide strand versus the lesser arm. The viability numbers of the matching 6mer seeds according to the siRNA 6mer seed screen are given. The sequences of the mature 5p or 3p arms are boxed in blue and black, respectively. Toxic seeds are shown in red, and non-toxic ones in blue

these genes were strongly enriched in clusters involved in chromosome segregation, DNA replication, cell cycle regulation, and mitosis, typical for 6mer seed toxicity-induced cell death (Supplementary Data 5). We quantified 30 of the 102 SGs in HeyA8 cells treated with Doxo at different time points using an arrayed quantitative PCR (Supplementary Fig. 9c). Twenty-four of the 30 genes' mRNAs were significantly downregulated as early as 7 h after treatment with no further reduction beyond 15 h after treatment, suggesting that their repression was the cause of cell death rather than a consequence. A Metascape analysis of all RNA-Seq data of downregulated RNAs in response to the toxic siL3, si34a-5p$^{Seed}$, miR-34a-5p, and the three genotoxic drugs suggested a common mode of action (Supplementary Fig. 9d). The GO clusters that were most significantly downregulated in all data sets were again related to DNA repair, cell cycle, and mitosis as described before for cells undergoing DISE[24].

To test whether treatment of cells with genotoxic drugs results in loading the RISC with toxic miRNAs, HeyA8 cells were treated with Doxo for 0, 20, 40, and 80 h and all four Ago proteins were pulled down using a GW182 peptide[42]. Interestingly, while the amount of AGO2 pulled down was the same at all time points, the amount of bound miRNA-sized RNAs substantially increased with longer treatment times (Fig. 7a). This was most likely the result of an overall increase in total small RNAs in the treated cells (Fig. 7b). Alternatively, this could also be a result of cells dividing more slowly and a stable RISC. miR-34a/b/c-5p bound to Ago proteins were upregulated at all time points (Supplementary Fig. 10a). To determine the contribution of miR-34a-5p and other miRNAs to the toxicity seen in cells exposed to the genotoxic drugs, we treated *Drosha* k.o. cells—devoid of most canonical miRNAs[43]—with the three genotoxic drugs (Supplementary Fig. 11a). These cells were hypersensitive to the toxicity induced by any of the three drugs. We attributed this response to the absence of most canonical miRNAs that protect cells from toxic RNAi-active sequences[20]. This result also suggested involvement of small RNAs that do not require Drosha for processing. As expected, the composition of small RNAs bound to Ago proteins dramatically varied between wild type and *Drosha* k.o. cells (Fig. 7c). In the absence of most canonical miRNAs, miR-320a-3p, which was previously shown not to require Drosha for its biogenesis[43], represented more than 86% of all Ago-bound miRNAs. Similar to HeyA8 cells (see Supplementary Fig. 10a), Ago-bound miR-34a-5p was upregulated in wild type but not in *Drosha* k.o. HCT116 cells upon Doxo treatment (Supplementary Fig. 10b, right). Interestingly, the average 6mer seed toxicity of all Ago-bound miRNAs >1.5-fold upregulated in HCT116 wt cells was significantly higher than the ones >1.5-fold downregulated in cells treated with Doxo (Fig. 7d, left). While in the *Drosha* k.o. cells, a number of non-toxic miRNAs were downregulated, the

only miRNA that was upregulated in the RISC (1.49-fold) was miR-320a-3p (Fig. 7d, right). However, upon closer inspection it became clear that this form of miR-320a-3p was shortened by two nucleotides at the 5′ end. This resulted in the shift of the 6mer seed into a G-rich sequence (Fig. 7d, right), converting a moderately toxic miRNA (average viability = 49.2%) into a highly toxic one (average viability = 9.3%). To test this predicted increase in toxicity experimentally, we transfected HeyA8 cells with either the authentic pre-miR-320a-3p or a miR-320a-3p duplex that corresponded to the shifted Ago-bound miR-320a-3p sequence (miR-320a-3p$^{Ago}$) (Fig. 7e). While pre-miR-320a-3p was not toxic, miR-320a-3p$^{Ago}$ completely blocked the growth of the cells. Toxicity of miR-320a-3p$^{Ago}$ was established in the four human and mouse cell lines (Fig. 7f). These data suggested that in the absence of other miRNAs that could kill cells through 6mer seed toxicity, miR-320a-3p (and possibly other small RNAs) may represent an alternative mechanism that ensures that genotoxic stressors can kill cells with defective miRNA processing often observed in cancer[44,45]. To test whether the 6mer seed toxicity exerted by miR-34a-5p would be synergistic with the toxicity caused by the three genotoxic drugs, we treated HeyA8 cells with a low dose (1 nM) of miR-34a-5p with low doses of either Doxo, Eto or Carbo (Supplementary Fig. 11b). No synergism was observed consistent with the assumption that genotoxic drugs are killing the cells at least in part through the use of toxic RNAi-active RNAs. In summary, our data suggest that certain tumor-suppressive miRNAs, such as miR-34a-5p and miR-320a-3p, exert their tumor-suppressive activities by carrying toxic 6mer seed sequences that can kill cancer cells by targeting SGs in C-rich regions close to the start of their 3′UTR. This activity may contribute to the cell death induced by genotoxic drugs.

## Discussion
We previously discovered a fundamental cell type- and species-independent form of toxicity that is evoked by the 6mer seed sequence in si/shRNAs that function similar to miRNAs[20]. We have now performed an siRNA screen that effectively tested the miRNA activities of all 4096 different 6mer seed sequences. Performing the screen in four cell lines (two human and two mouse) ensured that the results were relatively independent of species or cell type specific transcriptomes. The screen has discovered the rules underlying this seed toxicity and allows prediction of the 6mer seed toxicity for any siRNA, shRNA, miRNA with a known 6mer seed (https://6merdb.org).

Based on this screen, the toxicity of a number of tumor-suppressive miRNAs could be predicted solely on the basis of their 6mer seed sequences. The enrichment of G in the first 2–3 positions of the most toxic seeds is consistent with the way Ago

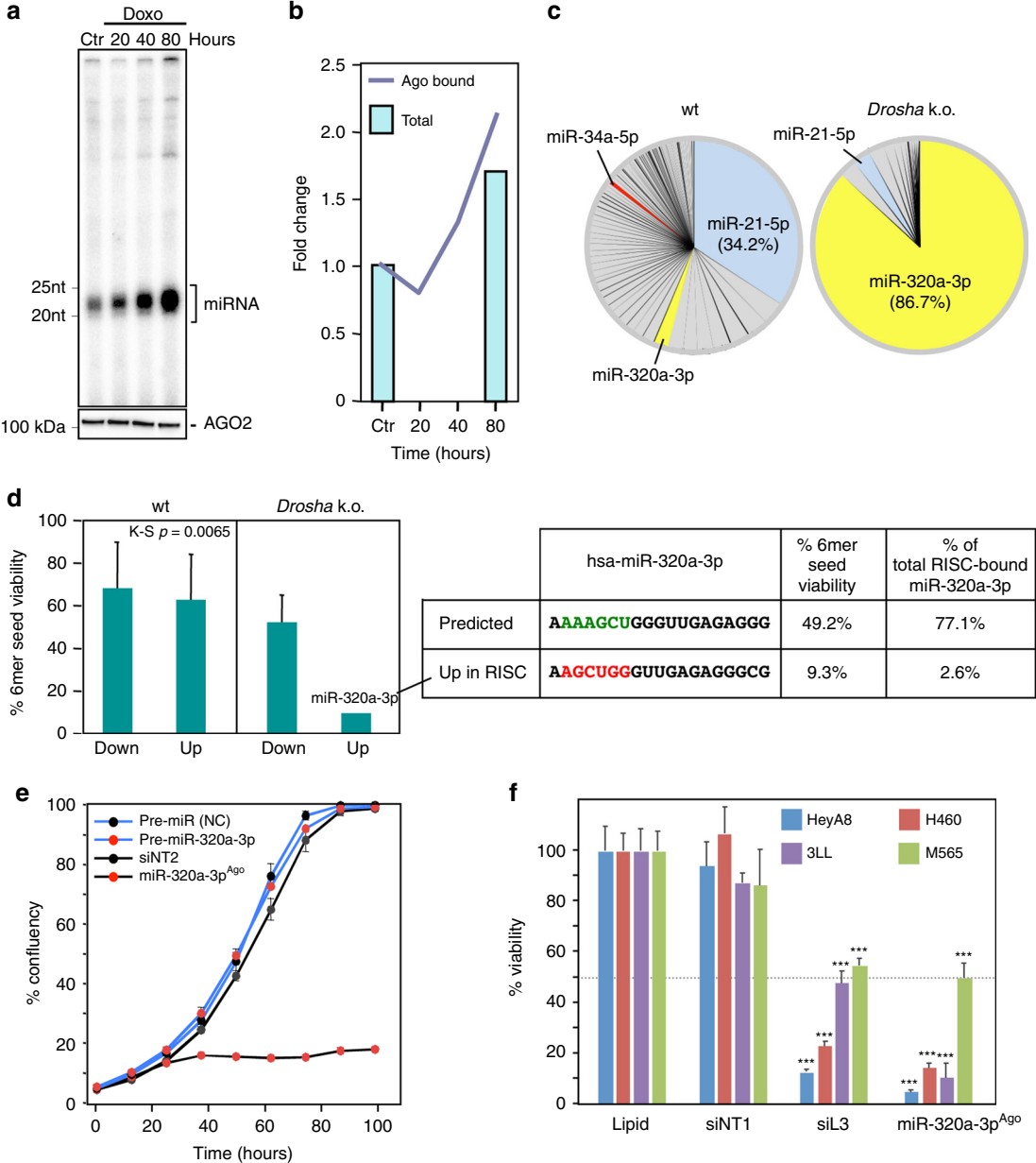

**Fig. 7** Genotoxic drugs cause upregulation of tumor-suppressive miRNAs with toxic 6mer seeds. **a** Top: Autoradiograph of radiolabeled RNAs pulled down with the Ago proteins from HeyA8 cells treated with doxorubicin (Doxo) for different times. Bottom: Western blot for the pulled down AGO2 of the same samples shown above. The images are representative of two biological duplicates. **b** Fold change of the total reads of Ago-bound small RNAs after 20, 40, or 80 h of Doxo treatment compared to the control sample from Ago-IP sequencing data (Ago-bound). Fold change of the total reads of cytosolic small RNAs in HeyA8 cells treated with Doxo for 80 h compared to the control sample from small RNA-Seq data is given (Total). Data are the combination of biological duplicates. **c** Pie charts showing the composition of miRNAs bound to Ago proteins after 50 h Doxo treatment in HCT116 wild type (left) or *Drosha* k.o. cells (right). **d** Left, 6mer seed viability (average between HeyA8 and H460 cells, two replicates) of the Ago-bound miRNAs most up- and downregulated in wt or *Drosha* k.o. cells after Doxo treatment. K–S test was used to calculate *p*-value. Right, Comparison of the predicted (and most abundant) sequence of miR-320a-3p and Ago-bound sequence of miR-320a-3p and their average viability found most upregulated in *Drosha* k.o. cells after Doxo treatment. Shown is variance of two biological replicates. **e** Percent cell confluence over time of HeyA8 cells transfected with 5 nM of controls, pre-miR-320a-3p, or an siRNA duplex that corresponds to the shifted form of miR-320a-3p (si-miR-320a-3p$^{Ago}$) that was found to be upregulated and bound to Ago proteins upon Doxo treatment. Data are representative of two independent experiments. Each data point represents mean ± SE of four replicates. **f** Viability changes (ATP content) in four cell lines 96 h after transfection with Lipid only, 10 nM of siNT1, siL3, a non-targeting pre-miR, or miR-320a-3p$^{Ago}$—the only shared upregulated miRNA in HeyA8 cells, HCT116 wild-type, and HCT116 *Drosha* k.o. cells—after Doxo treatment. *p*-Values were determined using Student's *t*-test. ***$p < 0.0001$. Samples were performed in triplicate (siNT1, siL3), six repeats (miR-320a-3p$^{Ago}$) and eight repeats (lipid)

proteins scan mRNAs as targets. This involves mainly the first few nucleotides (positions 1–3) of the seed[46]. miR-34a-5p contains two Gs in positions 1 and 2 of its 6mer seed. While miR-34a-5p is considered a master tumor-suppressive miRNA, no

single target has been identified to be responsible for this activity. Over 700 targets implicated in cancer cell proliferation, survival, and resistance to therapy have been described[16]. Our data now suggest that miR-34a-5p uses 6mer seed toxicity to target

hundreds of housekeeping genes. They provide the means to rationally design new artificial miRNAs as anticancer reagents that attack networks of SGs. In humans, miR-34a is highly expressed in many tissues. Consistent with our data that delivering siRNAs with toxic 6mer seeds to mice are not toxic to normal cells[21] miR-34a exhibits low toxicity to normal cells in vitro and in vivo[47]. miR-34a (MRX34) became the first miRNA to be tested in a phase I clinical trial of unresectable primary liver cancer[27,48]. The study was recently terminated and reported immune-related adverse effects in several individuals. It was suggested that these adverse effects may have been caused by either a reaction to the liposome-based carrier or the use of double-stranded RNA[16]. In addition, they may be due to an undesired gene modulation by miR-34a itself defined by sequences outside the 6mer seed[16]. Our data suggest that miR-34a exerts toxicity mostly through the 6mer seed of its 5p arm and that its 700 known targets may be part of the network of SGs that are targeted. The comparison of the RNA-Seq data of cells treated with either miR-34a-5p or si34a-5p$^{Seed}$ now allows to determine whether these two activities can be separated.

Our data provide evidence that genotoxic drugs kill cancer cells, at least in part, by triggering the toxic 6mer seed mechanism. Exposure of cancer cells to such drugs resulted in upregulation of tumor-suppressive miRNAs, most prominently of the p53 regulated miR-34 family[19]. While one report demonstrated that inhibiting miR-34a rendered cancer cells more resistant to cell death induced by genotoxic stress[49], another one found no effect of knocking out miR-34a on the sensitivity of HCT116 or MCF-7 cells to Doxo[50]. This mechanism may be highly redundant and may involve many miRNAs. Our analysis of Ago-bound miRNAs in Drosha k.o. cells suggest that in the absence of miR-34a, the noncanonical miR-320a-3p which was recently also found to be p53 regulated[51] may act as a backup miRNA that can still respond to genotoxic stress in case the amounts of other miRNAs are reduced, for instance in cases of mutations in miRNA biogenesis-associated genes frequently found in human cancers[44]. In addition, our recent data suggest that other toxic small RNAs can also be taken up by the RISC and negatively regulate cell growth through their toxic 6mer seed[23].

It was shown before that miRNAs overall avoid seed sequences that target the 3′UTR of survival/housekeeping genes[52,53]. Survival genes therefore are depleted in seed matches for the most abundant miRNAs in a cell. That also means 3′UTRs of SGs must be enriched in sequences not targeted by the seeds present in most miRNAs. Our combined data now suggest it is these sequences that toxic siRNAs and tumor-suppressive miRNAs with toxic 6mer seeds are targeting. Our analyses also suggest that most miRNAs have evolved over the last 800 million years by gradually depleting G in their seeds beginning at the 5′ end. In addition, the most abundant miRNAs have evolved to use the arm with the lower 6mer seed toxicity as the active guide strand, presumably to avoid killing cells. Only in a minority of tumor-suppressive miRNAs does the dominant guide strand contain a toxic seed. By ranking miRNAs according to whether they express the arm with the seed of higher toxicity, it is now possible to identify novel tumor-suppressive miRNAs (see https://6merdb.org).

In summary, we have determined the rules of RNAi targeting by toxic 6mer seeds. These rules allowed us to predict with some confidence which si/shRNAs or miRNAs have the potential to kill cells through their toxic 6mer seed. Toxic miRNAs seem to be involved in killing cancer cells in response to genotoxic drugs. Toxic 6mer seeds are present in a number of tumor-suppressive miRNAs that can kill cancer cells. Our data allow new insights into the evolution of miRNAs and provide evidence that 6mer seed toxicity is shaping the miRNA repertoire. In addition, they

now allow to develop super toxic artificial miRNAs for the treatment of cancer.

## Methods

**Reagents, cell lines, and antibodies.** HeyA8 (RRID:CVCL_8878) and H460 (ATCC HTB-177) cells were cultured in RPMI1640 medium (Cellgro Cat#10-040) supplemented with 10% fetal bovine serum (FBS) (Sigma Cat#14009 C) and 1% L-glutamine (Corning Cat#25-005). 3LL cells (RRID:CVCL_5653) were cultured in Dulbecco's modified Eagle's medium (DMEM) (Gibco Cat#12430054) supplemented with 10% FBS and 1% L-glutamine. Mouse hepatocellular carcinoma cells M565 were from a spontaneous formed liver cancer in a male mouse carrying a floxed Fas allele[54] and cultured in DMEM/F12 (Gibco Cat#11330) supplemented with 10% FBS, 1% L-glutamine and ITS (Corning #25-800-CR).

HCT116 parental (Cat#HC19023, RRID:CVCL_0291) and the Drosha k.o. clone (clone #40, Cat#HC19020) were purchased from Korean Collection for Type Cultures (KCTC). Both HCT116 cell lines were cultured in McCoy's 5A medium (ATCC, Cat#30-2007) supplemented with 10% FBS and 1% L-glutamine. All cell lines were authenticated by STR profiling. Anti-Argonaute-2 antibody (cat#ab186733, 1:1200) was purchased from Abcam, anti-β-actin antibody from Santa Cruz (#sc-47778, 1:5000), and secondary antibody for western blot was Goat anti-rabbit, IgG-HRP from Southern Biotech (#SB-4030-05, 1:5000). Etoposide (Cat#BML-GR307-0100) was purchased from Enzo Life Sciences; propidium iodide (#P4864) doxorubicin (Cat#D1515) and carboplatin (Cat#C2538) were from Sigma-Aldrich.

**siRNA screens and cell viability assay.** To design the non-toxic siRNA backbone used in the 4096 screen, the siNT2 sequence was used as a starting point and four positions in the center of siNT2 were replaced with the complementary nucleotides in order to remove any identity between the backbone siRNA and the toxic siL3 while retaining the same GC content. Two OMe groups were added to positions 1 and 2 of the passenger strand to prevent loading into the RISC. The 6mer seed region (position 2–7 on the guide strand) was then replaced with one of the 4096 possible seeds. Transfection efficiency was optimized for each of the four cell lines individually. RNA duplexes were first diluted with Opti-MEM to make 30 µl solution of 10 nM as final concentration in a 384-well plate by Multidrop Combi. Lipofectamine RNAiMAX (Invitrogen) was diluted in Opti-MEM (6 µl lipid + 994 µl of Opti-MEM for HeyA8, 15.2 µl lipid + 984.8 µl of Opti-MEM for M565, 9.3 µl of lipid + 990.7 µl of Opti-MEM for 3LL, and 7.3 µl of lipid + 993.7 µl of Opti-MEM for H460). After incubating at room temperature for 5–10 min, 30 µl of the diluted lipid was dispensed into each well of the plate that contains RNA duplexes. The mixture was pipetted up and down three times by PerkinElmer EP3, incubated at room temperature for at least 20 min, and then the mixture was mixed again by PerkinElmer EP3. Fifteen microliters of the mixture was then transferred into wells of three new plates (triplicates) using the PerkinElmer EP3. Fifty microliters cell suspension containing 320 HeyA8 or 820 M565 or 150 3LL or 420 H460 cells was then added to each well containing the duplex and lipid mix, which resulted in a final volume of 65 µl. Plates were left at room temperature for 30 min and then moved to a 37 °C incubator. Ninety-six hours post transfection, cell viability was assayed using CellTiter-Glo (Promega) quantifying cellular ATP content. Thirty-five microliters of medium were removed from each well, and 30 µl CellTiter-Glo cell viability reagent was added. The plates were shaken for 5 min and incubated at room temperature for 15 min. Luminescence was then read on the BioTek Synergy Neo2. The 4096 6mer seed-containing duplexes were screened in three sets for each cell line. Each set was comprised of five 384-well plates. A number of control siRNAs of known toxicity (including siNT1 and siL3) was added to each plate to compare reproducibility. All samples were set up in triplicate (on three different plates = 15 plates/set). The data in the HeyA8, H460, and M565 screens were normalized to lipid only on each plate. The 3LL screen which showed some drift between the sets was normalized to the average viability of the cells to siNT1 correcting the variability between sets.

**Transfection with short oligonucleotides.** For IncuCyte experiments, HeyA8 cells were plated in 50 µl antibiotic-free medium in a 96-well plate at 1000 cells/well, and 50 µl transfection mix with 0.1 µl RNAiMAX and siRNAs or miRNA precursors were added during the plating. For the AGO2 knockdown experiment, 100,000 cells/well HeyA8 cells were reverse transfected in six-well plates with either non-targeting (Dharmacon, cat#D-001810-10-05) or an AGO2-targeting siRNA SMARTpool (Dharmacon, cat#L004639-00- 005) at 25 nM. One microliter RNAiMAX per well was used for HeyA8 cells. Twenty-four hours after transfection with the SMARTpools, cells were reversed transfected in a 96-well plate with siNT2, si2733, or si2733 (see Supplementary Data 1) at 10 nM and monitored in the IncuCyte Zoom. To measure the knockdown efficiency, cells were lysed in RIPA buffer for western blot analysis 48 h after transfection with the SMARTpools.

All custom siRNA oligonucleotides were ordered from integrated DNA technologies (IDT) and annealed according to the manufacturer's instructions. In addition to the 4096 siRNAs of the screen the following siRNA sequences were used:

siNT1 sense: rUrGrGrUrUrUrArCrArUrGrUrCrGrArCrUrArATT;
siNT1 antisense: rUrUrArGrUrCrGrArCrArUrGrUrArArArCrCrAAA;

siNT2 sense: rUrGrGrUrUrUrArCrArUrGrUrUrGrGrUrGrGrUrGrATT;
siNT2 antisense: rUrCrArCrCrArCrArCrArUrGrUrArArArCrCrAAA;
siL3 sense: rGrCrCrUrUrCrArArUrUrArCrCrArUrArUTT;
siL3 antisense: rArUrArUrGrGrUrArArUrUrGrArArGrGrCAA;
si34a-5p^Seed sense: mUmGrUrUrUrArCrArUrGrUrUrGrCrUrUrTT;
si34a-5p^Seed antisense: rUrGrCrArGrUrArCrArUrGrUrArArArCrCrAAA;
miR-320a-3p^Ago sense: mCmGrCrCrUrUrCrUrCrArCrCrCrArGrCrUrTT
miR-320a-3p^Ago antisense:
rArArGrCrUrGrGrGrUrUrGrArGrArGrGrCrGAA.

The following miRNA precursors and negative controls were used: hsa-miR-34a-5p (Ambion, Cat. No# PM11030), hsa-let-7a-5p (Ambion, Cat. No# PM10050), hsa-miR-320a-3p (Ambion, Cat. No# PM11621), hsa-miR-15a-5p (Ambion, Cat. No# PM10235), and miRNA precursor negative control #1 (Ambion, Cat. No# AM17110).

**Western blot analysis**. Protein extracts were collected by lysing cells with RIPA lysis buffer (1% sodium dodecyl sulfate (SDS), 1% Triton X-100, 1% deoxycholic acid). Protein concentration was quantified using the DC Protein Assay kit (Bio-Rad, Hercules, CA). Thirty micrograms of protein were resolved on 8–12% SDS-polyacrylamide gel electrophoresis (PAGE) gels and transferred onto nitrocellulose membranes (Protran, Whatman) overnight at 25 mA. Membranes were incubated with blocking buffer (5% non-fat milk powder in 0.1% TBS/Tween-20) for 1 h at room temperature. Membranes were then incubated with the primary antibody diluted in blocking buffer overnight at 4 °C. Membranes were washed three times with 0.1% TBS/Tween-20. Secondary antibodies were diluted in blocking buffer and applied to membranes for 1 h at room temperature. After three more additional washes, detection was performed using the ECL reagent (Amersham Pharmacia Biotech) and visualized with the chemiluminescence imager G:BOX Chemi XT4 (Synoptics). All uncropped western blots are shown in Supplementary Figure 12.

**Monitoring cell growth by IncuCyte and cell death assays**. Cells were seeded between 1000 and 3000 per well in a 96-well plate in triplicates. The plate was then scanned using the IncuCyte ZOOM live-cell imaging system (Essen BioScience). Images were captured every 4 h using a ×10 objective. Cell confluence was calculated using the IncuCyte ZOOM software (version 2015A). For treatment with genotoxic drugs HeyA8 cells were seeded at 750 cells/well and HCT116 cells were seeded at 3000 cells/well in a 96-well plate and treated with one of the three genotoxic drugs (carboplatin, doxorubicin, or etoposide) at various concentrations at the time of plating. Solvent-treated (0.025% DMSO in medium) cells were used as control for etoposide. Medium-treated cells were used as control for carboplatin and doxorubicin. To assess cell viability, treated cells were subjected to a quantification of nuclear fragmentation or ATP content. To measure the level of nuclear fragmentation, cell pellet (500,000 cells) was resuspended in 0.1% sodium citrate, pH 7.4, 0.05% Triton X-100, and 50 μg/ml propidium iodide. After resuspension, cells were incubated 2–4 h in the dark at 4 °C. The percent of subG1 nuclei (fragmented DNA) was determined by flow cytometry. To measure the cellular ATP content, cells were reverse transfected with siRNAs in a 96-well plate at 1000 cells per well. Ninety-six hours after transfection, media in each well was replaced with 70 μl of fresh media and 70 μl of CellTiter-Glo cell viability reagent (Promega). The plates were shaken for 5 min and incubated at room temperature for 15 min. Luminescence was then read on the BioTek Cytation 5.

**RNA-Seq analysis**. For RNA-Seq data in Fig. 5a, 50,000 cells/well HeyA8 cells were reversed transfected in duplicate in six-well plates with 10 nM of either pre-miR-34a-5p or si-miR-34a-5p^Seed with their respective controls. The transfection mix was replaced 24 h after transfection. Cells were lysed 48 h after transfection using Qiazol. For the RNA-Seq data in Supplementary Fig. 9, HeyA8 cells were seeded at 50,000 cells per well in a six-well plate and treated with three genotoxic drugs in duplicate: carboplatin (25 μg/ml), doxorubicin (50 ng/ml), and etoposide (500 nM). Medium-treated cells were used as control for carboplatin and doxorubicin-treated cells. Solvent control-treated cells (0.025% DMSO in medium) were used as control for etoposide. Cells were lysed after 80 h drug incubation using Qiazol. Total RNA was isolated using the miRNeasy Mini Kit (Qiagen, Cat. No# 74004) following the manufacturer's instructions. An on-column digestion step using the RNAse-free DNAse Set (Qiagen, Cat.No# 79254) was included for all RNA-Seq samples. RNA libraries were generated and sequenced (Genomics Core facility at the University of Chicago). The quality and quantity of the RNA samples were checked using an Agilent bio-analyzer. Paired end RNA-SEQ libraries were generated using Illumina TruSEQ TotalRNA kits using the Illumina provided protocol (including a RiboZero rRNA removal step). Small RNA-SEQ libraries were generated using Illumina small RNA SEQ kits using the Illumina provided protocol. Two types of small RNA-SEQ sub-libraries were generated: one containing library fragments 140–150 bp in size and one containing library fragments 150–200 bp in size (both including the sequencing adaptor of about 130 bp). All three types of libraries (one RNA-SEQ and two small RNA-SEQ) were sequenced on an Illumina HiSEQ4000 using Illumina provided reagents and protocols. Adaptor sequences were removed from sequenced reads using TrimGalore (https://www.bioinformatics.babraham.ac.uk/projects/trim_galore). The trimmed reads were aligned to the hg38 version of the human genome, using either Tophat v2.1.0

(RNA-Seq data in Supplementary Fig. 9) or STAR v2.5.2 (RNA-Seq data in Fig. 5a). In either case, aligned reads were associated with genes using HTSeq v0.6.1, and the UCSC hg38 transcriptome annotation from iGenomes. Differentially expressed genes were identified using the edgeR R package.

**Arrayed real-time PCR**. The top 30 most downregulated SGs shared among HeyA8 cells treated with carboplatin, doxorubicin, and etoposide based on the RNA-Seq analysis were selected for a kinetics analysis using real-time PCR. To prepare the RNAs for the kinetics analysis, 75,000 HeyA8 cells were seeded in 15 cm plates. Twenty-four hours after plating, one plate of HeyA8 cells were lysed in QIAzol as the control sample. The rest of the plates were treated with 50 ng/ml doxorubicin for 7, 14.5, and 21 h, respectively before being lysed in QIAzol. To perform the arrayed real-time PCR, 200 ng total RNA per sample was used as the input to make cDNA using the high-capacity cDNA reverse Transcription Kit (Applied Biosystems #4368814). For TaqMan Low Density Array (TLDA) profiling, custom-designed 384-well TLDA cards (Applied Biosystems, Cat. No#4346799) were selected and used according to the manufacturer's protocols. For each sample, 20 μl cDNA was mixed with 80 μl water and 100 μl TaqMan Universal PCR Master Mix (Applied Biosystems, Cat. No#4304437). A total volume of 100 μl of each sample was loaded into the eight loading ports on the TLDA card (2 ports for each sample, 4 samples total on one card). The qPCR assays used to detect the 30 genes on the TLDA card are as follows: HIST1H2AI (Hs00361878_s1), CENPA (Hs00156455_m1), HJURP (Hs00251144_m1), FAM72D (Hs00416746_m1), CCNA2 (Hs00996788_m1), KIF20A (Hs00993573_m1), PRC1 (Hs01597839_m1), KIF15 (Hs01085295_m1), BUB1B (Hs01084828_m1), SCD (Hs01682761_m1), AURKA (Hs01582072_m1), NUF2 (Hs00230097_m1), NCAPH (Hs01010752_m1), SPC24 (Hs00699347_m1), KIF11 (Hs00189698_m1), TTK (Hs01009870_m1), PLK4 (Hs00179514_m1), AURKB (Hs00945858_g1), CEP55 (Hs01070181_m1), HMGCS1 (Hs00940429_m1), TOP2A (Hs01032137_m1), KIF23 (Hs00370852_m1), INCENP (Hs00934447_m1), CDK1 (Hs00938777_m1), HIST2H2BE (Hs00269023_s1), KNL1 (Hs00538241_m1), NCAPD2 (Hs00274505_m1), RACGAP1 (Hs01100049_mH), SPAG5 (Hs00197708_m1), KNTC1 (Hs00938554_m1). GAPDH (Hs99999905_m1) was used as the endogenous control. qPCR assay for individual gene was done in technical triplicates on each TLDA card. Statistical analysis was performed using Student's t-test.

**Ago affinity peptide purification**. To purify the FLAG-GST-T6B WT and mutant, constructs were expressed in BL21-Gold(DE3)pLysS-competent cells (Agilent). Bacteria, induced with 1 mM isopropyl β-D-1-thiogalactopyranoside (IPTG), were grown in 1 liter overnight at 18 °C to OD 0.6. The bacteria were sedimented at 4000 g for 15 min and resuspended in 25 ml GST-A buffer (1 mM 4-(2-aminoethyl) benzenesulfonyl fluoride hydrochloride (AEBSF), 1 mM DTT in phosphate-buffered saline (PBS)) supplemented with 1 mg/ml lysozyme (Sigma). Samples were sonicated three times for 3 min at 100% amplitude (Sonics, VCX130) and cleared by centrifugation at 20,000 g for 20 min. The lysate was loaded onto a column containing 2 ml of bead volume glutathione Sepharose beads (Sigma) and washed two times with GST-A buffer. The GST-tagged protein was eluted in 10 ml of GST-B buffer (20 mM Tris, pH 8.0, and 10 mM glutathione in PBS). The peptide was concentrated using Amicon Ultra-15 Centrifugal Filter Unit (Millipore) and desalted using Zeba Spin Desalting Columns (ThermoFisher).

**Ago pull down and small RNA-seq**. HeyA8 ($5–7 × 10^6$), HCT116 wild type ($1.2–1.6 × 10^8$), or *Drosha* k.o. ($4.8–6.3 × 10^7$) cells treated with doxorubicin were lysed in NP40 lysis buffer (20 mM Tris, pH 7.5, 150 mM NaCl, 2 mM EDTA, 1% (v/v) NP40, supplemented with phosphatase inhibitors) on ice for 15 min. The lysate was sonicated three times for 30 s at 60% amplitude (Sonics, VCX130) and cleared by centrifugation at 12,000 g for 20 min. AGO1-4 were pulled down by using 500 μg of Flag-GST-T6B peptide[42] and with 60 μl anti-Flag M2 magnetic beads (Sigma-Aldrich) for 2 h at 4 °C. The pull down was washed three times in NP40 lysis buffer. During the last wash, 10% of beads were removed and incubated at 95 °C for 5 min in 2× SDS-PAGE sample buffer. Samples were run on a 4–12% SDS-PAGE and transferred onto nitrocellulose membrane. The pull-down efficiency was determined by immunoblotting against AGO2 (Abcam 32381). To the remaining beads 500 μl TRIzol reagent was added and the RNA extracted according to the manufacturer's instructions. The RNA pellet was diluted in 20 μl of water. The sample was split and half of the sample was dephosphorylated with 0.5 U/μl of CIP alkaline phosphatase at 37 °C for 15 min and subsequently radiolabeled with 0.5 μCi γ-32P-ATP and 1 U/μl of T4 PNK kinase for 20 min at 37 °C. The AGO1-4 interacting RNAs were visualized on a 15% Urea-PAGE. To prepare a small RNA library, RNA was ligated with 3′ adenylated adapters and separated on a 15% denaturing urea-PAGE. The RNA corresponding to insert size of 19–35 nt was eluted from the gel, ethanol precipitated followed by 5′ adapter ligation. The samples were separated on a 12% Urea-PAGE and extracted from the gel. Reverse transcription was performed using Superscript III reverse transcriptase and the cDNA amplified by PCR. The cDNA was sequenced on Illumina HiSeq 3000.
Adapter sequences:
Adapter 1—NNTGACTGTGGAATTCTCGGGTGCCAAGG;
Adapter 2—NNACACTCTGGAATTCTCGGGTGCCAAGG;

Adapter 3—NNACAGAGTGGAATTCTCGGGTGCCAAGG;
Adapter 4—NNGCGATATGGAATTCTCGGGTGCCAAGG;
Adapter 47—NNTCTGTGTGGAATTCTCGGGTGCCAAGG;
Adapter 48—NNCAGCATTGGAATTCTCGGGTGCCAAGG;
Adapter 49—NNATAGTATGGAATTCTCGGGTGCCAAGG;
Adapter 50—NNTCATAGTGGAATTCTCGGGTGCCAAGG.
RT primer sequence: GCCTTGGCACCCGAGAATTCCA;
PCR primer sequences:
CAAGCAGAAGACGGCATACGAGATCGTGATGTGACTGGA
GTTCCTTGGCACCCGAGAATTCCA.

**Data analyses**. GSEA was performed using the GSEA software version 3.0 from the Broad Institute downloaded from https://software.broadinstitute.org/gsea/. A ranked list was generated by sorting genes according the Log$_{10}$(fold down-regulation). The Pre-ranked function was used to perform GSEA using the ranked list. One thousand permutations were used. Default settings were used. The ~1800 SGs and ~420 non-SGs defined previously[20] were used as custom gene sets. Default settings were used.

The list of SG and expression-matched non-SGs were generated by taking the survival and expression-matched non-SGs used previously[20] and retaining only the 938 genes in each group of expression-matched survival and non-SGs with an average expression across all RNA-seq datasets above 1000 RPMs (see Supplementary Data 6).

Sylamer analysis[29] was used to find enrichment of small word motifs in the 3′UTRs of genes enriched in those that are most downregulated. 3′UTRs were used from Ensembl, version 76. As required by Sylamer, they were cleaned of low-complexity sequences and repetitive fragments using respectively Dust[55] with default parameters and the RSAT interface[56] to the Vmatch program, also run with default parameters. Sylamer (version 12-342) was run with the Markov correction parameter set to 4. Bonferroni-adjusted p-values were calculated by multiplying the unadjusted p-values by the number of permutations for each length of word searched for.

The GO enrichment analyses shown in Fig. 5b and Supplementary Fig. 6b were performed using the GOrilla GO analysis tool at http://cbl-gorilla.cs.technion.ac.il using default setting using different p-value cut-offs for each analysis. GO analysis in Supplementary Data 5 was done using DAVID 6.8 (https://david.ncifcrf.gov) using default settings. GO analyses across multiple data sets were performed using the software available on www.Metascape.org and default running parameters.

Density plots showing the contribution of the four nucleotides G, C, A, and U at each of the 6mer seed positions were generated using the Weblogo tool at http://weblogo.berkeley.edu/logo.cgi using the frequency plot setting.

Venn diagrams were generated using http://bioinformatics.psb.ugent.be/webtools/Venn/ using default settings.

The scatter plot in Fig. 5a was generated using R package ggplot2. 10875 genes with RPM > 1 (average RPM of the 8 RNA-seq samples) and adjusted p-value <0.05 were included. In all, 3696 genes were significantly upregulated in both mir-34a-5p and si34a-5p$^{Seed}$ treated samples. Four thousand two hundred and seven genes were significantly downregulated in both mir-34a-5p and si34a-5p$^{Seed}$ treated samples. Seven hundred and thirteen genes were only downregulated and 792 genes were only upregulated in si34a-5p$^{Seed}$-treated samples. Seven hundred and thirty genes were only downregulated and 737 genes were only upregulated in mir-34a-5p-treated samples. One hundred and ninety-three genes out of the total 10875 genes were omitted in the graph as the range for X and Y axes were set as −3 to 3.

**Identification of the most and least toxic 6mer seeds**. To identify the 20 and 100 most and least toxic seeds to both human cell lines all 4096 seeds were ranked for each cell line from highest to lowest toxicity. The 20 seeds with the highest toxicity to both HeyA8 and H460 cells were found in the top 46 most toxic seeds to both cells and the 20 seeds shared to be least toxic were found in the bottom 149 seeds in each ranked group. The 100 most and least toxic seeds for both cell lines were identified in the same way and all groups of seeds are shown in Supplementary Data 2.

**Metaplots of 6mer seed match locations**. 3′UTR sequences were downloaded from Ensembl Biomart. In order to reduce redundancy in the sequences, a single longest 3′UTR (and associated transcript) was chosen to represent each gene. A custom perl script (makeSeedBed.pl) was written to identify exact matches to all seeds (reverse complement) in all sequences, and to output the coordinates of those matches in bed file format. A custom R script (plotBedMetaPlot.R) was written that uses the GenomicRanges[57] and Sushi[58] R packages to calculate the coverage of seeds across all sequences in a given set, and to create a plot of that coverage. The custom scripts and the input data are available in the cloud-based computational reproducibility platform Code Ocean at https://doi.org/10.24433/CO.9a3eb292-6e89-44f0-b9b0-bdd779f97516.

**eCDF plots**. A custom perl script (annotateWithSeeds.pl) was written to identify exact seed matches (reverse complement) to all seeds in all sequences, and to output the total counts of the different types of seeds (generally toxic versus non-

toxic) in the sequences. To compare the presence of toxic and non-toxic seed matches in expression-matched survival and non-SGs, a custom R script (makeECDFplot.180615.R) was written that takes as input two different sets of genes (SGs and non-SGs) and the list of the counts of toxic and non-toxic seeds (reverse complement) in all genes, and plots the cumulative distribution function for the count statistics in each gene set. In Fig. 3b the ratio of the seed match counts to the 20 most and least toxic seeds in the 5′UTR, CDS, first 1000 bp of 3′UTR, and full 3′UTR (not shown) were compared between pairs of 938 expression-matched survival and non-SGs. In Supplementary Fig. 5a, this analysis was repeated with the 100 most and least toxic seeds to both human cell lines. The custom scripts and the input data are available in Code Ocean at https://doi.org/10.24433/CO.b755ec2b-00d8-4281-9fa1-2a484fd7521b. To determine the dependence of mRNAs regulation on miR-34a-5p seed presence in their 3′UTR, a custom R script (makeECDFplot.cetoData.R) was written that takes as input a list of gene sets and a table of logFC expression for those genes upon miR-34a-5p or si34a-5p$^{Seed}$ over-expression. This Rscript then plots the cumulative distribution function for the logFC expression data in each gene set. The custom scripts and the input data are available in Code Ocean at https://doi.org/10.24433/CO.31ec8deb-8282-4a90-98e6-b80a0ba881cb.

**Relation between miRNA seed conservation, age, and toxicity**. Information on miRNA seed family conservation and seed sequence were downloaded at http://www.targetscan.org/vert_71/ from TargetScan Human 7.1. The toxicity of each mature human miRNA arm sequence in the TargetScan dataset was assigned according to the average toxicity induced by the siRNA in HeyA8 and H460 siRNA screens harboring the identical 6mer seed sequence. A list of miRNA ages corresponding to ~1025 miRNA loci was acquired from ref. [31] and was calculated using a modified version of ProteinHistorian[31]. This list was used to assign ages to roughly ~1400 mature miRNA arms found in the TargetScan dataset.

TargetScan 7.1 partitions the seed family conservation into four groups: highly conserved (group #2), conserved (group #1), low conservation but still annotated as a miRNA (group #0), and low conservation with the possibility of misannotation (group #-1). Probability density and eCDF plots for the assigned 6mer seed-dependent toxicities were generated for each seed family conservation group as defined by Targetscan (groups -1, 0, 1, and 2) using ggplot2 in R. Probability density and eCDF plots were also generated to show how young (<10 million years) and old (>800 million years) miRNAs compare in terms of the seed-dependent toxicity. All differences between groups in terms of seed-dependent toxicity (always the average of the toxicity determined in HeyA8 and H460 cells) were analyzed using a two-sample two-tailed Kolmogorov–Smirnov test in R.

**Assessment of dominant arm seed toxicity**. The expression (RPM) of miRNA 5p and 3p arms across 135 tissue samples was collected from MIRMINE[59]. A miRNA was considered expressed if the sum of the normalized reads for both arms was above 5 for each sample. A value of 0 was replaced with 0.01 to avoid a division by 0 error. A miRNA arm was considered the dominant species if its expression was at least 25% greater than the other arm per sample. The dominant arm for each miRNA across all samples was calculated by determining which arm was dominantly expressed in more samples (>50% of samples where the miRNA was considered expressed). The miRNAs that have only one annotated arm in miRBase were considered to have only one dominant arm. The seed toxicity values for each arm were extracted from the 4096 siRNA screen data (% average viability for the human HeyA8 and H460 cells) and used to calculate the ratio between the dominant arm's toxicity and the lesser arm's toxicity; miRNAs that only had one expressed arm were not considered in the analyses shown in Fig. 6d but are all included on the website: 6merdb.org.

To compare the 6mer seed toxicity between upregulated and downregulated Ago-bound miRNA populations in HCT116 cells after Doxo treatment we analyzed the reads obtained from RNA-Seq analysis of Ago-bound RNAs. After removing the reads that were either shorter than 19 nt or longer than 26 nt in length, the reads were blasted against a miRNA database consisting of all human miRNA mature sequence information obtained from miRBase. A threshold of 100% identity for an at least 16 nt long stretch without any gaps was set for the BLAST analysis. After discarding sequences with no significant BLAST result, the remaining sequences were trimmed from the 3′ end so that all reads were now 19nt in length. This was done to determine for each miRNA the relevant 5′ end to obtain the 6mer seed sequence (position 2–7). All the reads that shared the same 5′ sequence and miRNA names were collapsed while adding up the number of reads of each sequence in each condition. To compare the 6mer seed toxicity between up- and downregulated miRNAs, we calculated the average 6mer seed toxicity for miRNA sequences that were either 1.5-fold up or 1.5-fold downregulated in Doxo-treated wt samples compared to medium-treated control samples (after removing sequences that had less than 100 collapsed reads in Doxo-treated wt samples). In each group miRNAs were ranked according to highest base mean expression and groups were compared (Fig. 7d). Statistically significance was determined using the K-S test. The comparison was repeated for Drosha k.o. cells where the 5′ shifted form of miR-320a-3p was the only miRNA found to be upregulated (1.5-fold).

**Analysis of miRtron and non-miRtron groups**. MiRtrons and non-miRtrons were recently reported[60] and consisted of miRNAs that are listed in miRBase v21 as expressing both arms. Comparing 6mer toxicity of miRtrons and non-miRtrons from this list was done as described above for young/old non-conserved/conserved miRNAs. Both arms were considered. Probability density (Fig. 6c) and eCDF plots (Supplementary Fig. 7c) were generated to show how miRtrons and non-miRtrons compare in terms of the seed-dependent toxicity. To calculate the 6mer toxicity across the entire miRtron sequences, we extracted all possible 6-nt stretches from the first 17 nts of the 428 mature miRtron sequences beginning at the 5′ end using a 6-nt sliding window (12 different start positions in total). The first 17 nts were chosen because all miRtrons sequences are 17–25 nt in length. Average 6mer toxicity of the 428 miRtrons was calculated for each start position and plotted in Supplementary Fig. 7e. To visualize the nucleotide content across all miRtrons, the first eight nucleotides from the 5′ end and the last eight nucleotides from the 3′ end were extracted from the 428 mature miRtron sequences and analyzed using the Weblogo tool.

**Seed viability of shRNAs derived from the CD95L sequence**. An RNAi lethality screen composed of every shRNA sequence that can be derived from the CD95L CDS was conducted previously[20]. In this screen, toxicity of each shRNA was assessed in two ways: (1) fold underrepresentation of the shRNA after infection with the shRNA-expressing lentivirus compared to its representation in the plasmid pool and (2) fold representation of the shRNA after infection and treatment with doxycycline compared to cells that were infected but did not receive doxycycline. The first analysis allowed us to quantify toxicity associated with leaky shRNA expression. The second analysis quantified toxicity associated with strong shRNA expression following treatment with doxycycline.

For each shRNA, the average fold downregulation was calculated from both of these toxicity assessments. Then, the seed sequence of each shRNA was extracted and assigned an average viability score, which was a composite of the % viabilities determined in the 4096 siRNA arrayed screen for both HeyA8 and H460 cells.

Pearson's correlation was determined for each CD95L-derived shRNA between its associated fold downregulation in the shRNA screen[20] and the average seed sequence viability determined in the 4096 siRNA screen.

In addition, the CD95L shRNAs were split into two groups: (1) 137 shRNAs with an average fold downregulation (as determined in the shRNA lethality screen) above 5 and (2) a control group with a matching number of shRNAs whose fold deregulation had an absolute value closest to 0. The average seed viability (as determined from the siRNA screen) was extracted for the shRNAs in these two groups and compared using the two-sample, two-tailed rank-sum test.

**Statistical analyses**. Continuous data were summarized as means and standard deviations (except for all IncuCyte experiments where standard errors are shown) and dichotomous data as proportions. Continuous data were compared using t-tests for two independent groups and one-way ANOVA for three or more groups. For evaluation of continuous outcomes over time, two-way ANOVA was performed using the Stata 14 software with one factor for the treatment conditions of primary interest and a second factor for time treated as a categorical variable to allow for non-linearity. Comparisons of single proportions to hypothesized null values were evaluated using binomial tests. Statistical tests of two independent proportions were used to compare dichotomous observations across groups. Pearson correlation coefficients (r) and p-values as well as Wilcoxon rank test were calculated using StatPlus (v. 6.3.0.5). Kolmogorov–Smirnov (K-S) two-sample two-sided test was used to compare different probability distributions shown in all density plots and eCDF plots. Wilcoxon rank-sum test was used to test for statistical significance in the analysis of the toxicity all miRNA arms in Fig. 6d. The Fisher Exact test was used to calculate p-values in Fig. 6b to determine whether the percent frequency of G versus non-G (A, C, and U) nucleotides at each position along the 6mer seed was different between young (<10 million years) and ancient (>800 million years) miRNAs. The effects of treatment versus control over time were compared for Drosha k.o. and wild-type cells by fitting regression models that included linear and quadratic terms for value over time, main effects for treatment and cell type, and two- and three-way interactions for treatment, cell type, and time. The three-way interaction on the polynomial terms with treatment and cell type was evaluated for statistical significance since this represents the difference in treatment effects over the course of the experiment for the varying cell types.

## Data availability

RNA sequencing data generated for this study is available in the GEO repository: GSE111379 and GSE111363. All 6mer seed toxicity data of the 4096 siRNA screen in HeyA8, H460, M565, and 3LL cells are available in searchable form at https://6merdb.org. The data that support the findings of this study are available from the corresponding author upon request.

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

## Acknowledgements

We are indebted to Dr. Leon Platanias for his generous support of the seed screen and to Denise Scholtens for help with biostatistics. M.H. was supported by the Intramural Research Program of NIAMS and A.A.S. by the Swedish Research Council postdoctoral fellowship. This work was funded by training grant T32CA009560 (to W.E.P.), R35CA197450 (to M.E.P.), and R50CA221848 (to E.T.B.).

## Author contributions

M.E.P., Q.Q.G., W.E.P. and A.E.M. conceived the study. Q.Q.G and W.E.P. performed the majority of the experiments and data analysis. S.C. performed the 6mer seed screens. A.A.S. performed the Ago pull-down experiments. E.T.B. performed data analyses. J.M.P. designed and implemented the website. M.H. provided critical conceptual input. M.E.P. designed the study, guided the interpretation of the results, and drafted the manuscript. All authors discussed the results, edited, and approved the draft and final versions of the manuscript.

## Additional information

**Competing interests:** The authors declare no competing interests.

