## [Peer Review File · Nature Communications]

REVIEWERS' COMMENTS:

Reviewer #3 (Remarks to the Author):

The original manuscript contained a section on clinical correlation using ovarian cancer samples from TCGA. These sections have now been removed, and the manuscript is now restricted to analysis of the toxic effects of 6mer seeds in cell line models. This reviewer is not an expert in this field, so I will allow the other reviewers to make comments.

Reviewer #4 (Remarks to the Author):

Review of "6mer seed toxicity in tumor suppressive microRNAs"

The authors re-submitted a manuscript in which they systematically analyze the effects of all 4096 possible 6-mer seed-containing siRNAs on 4 cancer cell lines. They find highly variable impacts on cell viability and identify G and GC content as the main driver of "seed toxicity" by an enhanced probability to target critical "survival genes". The concept of seed toxicity provides a new angle to look at miRNA function and evolution, and is of interest to cancer biology.

The present version of the manuscript is a major improvement over the initial submission with more data added, more controls, validation, and many main figures enhanced. This referee is particularly happy to see that many suggestions were picked up by the authors and proved useful to improve the figures and main text. The new manuscript raises a handful of new (minor) issues and questions, which should be straight-forward to address.

1) The new intro is much clearer.

2) Fig. 3B lacks x-axis label.

3) I.201 the majority of *differentially* expressed genes were down-regulated ? Or indeed > 78% of *all* expressed genes down?

4) Fig. 3D perhaps merge SG and nonSG plots and show $\log_2(\text{SG}/\text{nonSG})$ on y-axis? This might show the "area of reduced A/U content in SGs" more clearly. Error bars/envelopes from sub-sampling/boot-strapping? Any other idea for assessment of significance?

5) Fig. 3E is missing graphs (in two different PDF viewers): most toxic seed matches (upper panel), nonSGs (middle panel), and nonSGs (lower panel) are missing. I can see the corresponding data in the Supplementary Figures. Perhaps a transparency issue? Red + Green in same plot should be avoided. Consider showing log-ratio of SGs to nonSGs (see point 4).

6) Supplementary Figure S5:

red/green in same plot not ideal

looks like the major difference is elevated seed match density in 5'UTR, CDS, and first ~100nt of 3'UTR for toxic seeds. In comparison, SG vs. non-SG looks much weaker, esp in 3'UTR. Consider showing log-ratio of SGs to nonSGs (see point 4). Could the much stronger targeting of toxic seeds to CDS be an alternative mechanisms, by which toxic seeds perhaps interfere more broadly with translation? Perhaps a point for discussion.

7) Fig. 4F: please avoid red+green in same plot

8) Gradual depletion of G: significant? a out of b seeds have G1 in < 10Mya, c out of d have G1 in > 800Mya -> hypergeometric or Fisher's exact test

9) miRtron observation: what is the null expectation if 6mers were randomly sampled from miRtrons (or length-matched introns)?

10) miR-449 as backup for miR-34a. If the tumor suppressive function is being selected for, and many different seeds are toxic, why is the exact same seed used as backup? Why are there not more toxic-seed containing tumor suppressor miRNAs? Why are toxic seeds overall rare among endogenous miRNAs and depleted from ancient miRNAs, whereas many tumor suppressing proteins are extremely conserved? Their general toxicity would only be a problem if expression were not restricted to genotoxic stress situations. Perhaps here is another argument for the evolutionary result: old miRNAs tend to have much broader domains of expression...

11) Fig. S10a: "total read numbers" or "reads per million"? If indeed counts, then error-bars seem small for miR-34b-5p and miR-34c-5p and large for miR-34a-5p. "Variance of two biol. Replicates" should just be replaced by the two data points (swarm plot).

12) The discussion mentions miR-34a loss-of-function experiments with somewhat inconclusive results. Regarding gain-of-function: are cells treated with toxic seed siRNAs more sensitive to genotoxic drugs than cells treated with non-toxic seed siRNAs? Are there synergistic/sensitizing effects? Could this allow to disentangle whether toxic seeds (miR-34a) are downstream of genotoxic triggers or an independent enhancer of cell death? While absolutely not required, this experiment seems doable and could potentially add substantial weight to the oncology aspect of the paper.

Reviewer #4 (Remarks to the Author):

Review of "6mer seed toxicity in tumor suppressive microRNAs"

The authors re-submitted a manuscript in which they systematically analyze the effects of all 4096 possible 6-mer seed-containing siRNAs on 4 cancer cell lines. They find highly variable impacts on cell viability and identify G and GC content as the main driver of "seed toxicity" by an enhanced probability to target critical "survival genes". The concept of seed toxicity provides a new angle to look at miRNA function and evolution, and is of interest to cancer biology.

The present version of the manuscript is a major improvement over the initial submission with more data added, more controls, validation, and many main figures enhanced. This referee is particularly happy to see that many suggestions were picked up by the authors and proved useful to improve the figures and main text. The new manuscript raises a handful of new (minor) issues and questions, which should be straight-forward to address.

Response: ...and we are very grateful that this reviewer took the time to advise us in such great detail. We appreciate the very constructive and helpful comments. As a result the manuscript is indeed much improved.

2) Fig. 3B lacks x-axis label.

Response: This has been fixed

3) I.201 the majority of *differentially* expressed genes were down-regulated ? Or indeed > 78% of *all* expressed genes down?

Response: We have clarified this sentence.

4) Fig. 3D perhaps merge SG and nonSG plots and show $\log_2(\text{SG}/\text{nonSG})$ on y-axis? This might show the "area of reduced A/U content in SGs" more clearly. Error bars/envelopes from sub-sampling/bootstrapping? Any other idea for assessment of significance?

We have restructured this figure and have added statistics. We did test the \log_2 ratio as suggested but due to major differences in ratio in areas of low signal the noise was too high and we decided to stick with the current presentation. It provides more information. However, we have removed any red/green combinations.

5) Fig. 3E is missing graphs (in two different PDF viewers): most toxic seed matches (upper panel), nonSGs (middle panel), and nonSGs (lower panel) are missing. I can see the corresponding data in the Supplementary Figures. Perhaps a transparency issue? Red + Green in same plot should be avoided. Consider showing log-ratio of SGs to nonSGs (see point 4).

The reviewer is correct. The transparency did not covert in the pdf file. We apologize for this oversight.

We did test the \log_2 ratio as suggested but due to issues mentioned under 4 we decided to stick with this current presentation. We have removed red/green combinations.

6) Supplementary Figure S5:

red/green in same plot not ideal

looks like the major difference is elevated seed match density in 5'UTR, CDS, and first ~100nt of 3'UTR for toxic seeds. In comparison, SG vs. non-SG looks much weaker, esp in 3'UTR. Consider showing log-ratio of SGs to nonSGs (see point 4). Could the much stronger targeting of toxic seeds to CDS be an alternative mechanisms, by which toxic seeds perhaps interfere more broadly with translation? Perhaps a point for discussion.

The stronger occurrence of toxic seed matches in the CDS of all genes is intriguing. However, as there is no significant difference between SGs and nonSGs, We do not think it is relevant to the 6mer seed toxicity described here. We have removed any red/green combinations.

7) Fig. 4F: please avoid red+green in same plot

We have changed the color scheme in this figure.

8) Gradual depletion of G: significant? a out of b seeds have G1 in < 10Mya, c out of d have G1 in > 800Mya - > hypergeometric or Fisher's exact test

Response: We have performed a Fisher's exact test for each of the 6 seed positions comparing the youngest

to the oldest miRNAs. The change in seed composition is statistically significant for 3 of the positions. We believe that the difference in seed composition between positions 1 and 2 did not reach statistical significance because the oldest most conserved miRNAs are comprised of both a majority of nontoxic miRNAs and a minority of toxic tumor suppressive miRNAs such as miR-34a. The latter have Gs in their first two positions. We have added this information to the revised manuscript.

9) miRtron observation: what is the null expectation if 6mers were randomly sampled from miRtrons (or length-matched introns)?

Response: We have approached this question in two ways. 1) Rather than randomly sampling 6mers from miRtrons we decided to plot the average nucleotide content of all 428 miRtrons in our analysis. As miRtrons have different lengths (between 17 and 25nts) we are showing the analysis of the first 8 and the last 8 nucleotides across all miRtrons in new Figure S7d. Consistent with the G rich 6mer toxicity observed in our work the highest G content is indeed in positions 2-7, the 6mer seed. This effect is not just due to a different nucleotide composition at the start of an intron (although that can factor in) but likely to the specific function of miRtrons. We checked the miRtrons and they all start at different positions following the intron start site. 2) We are now plotting the average 6mer seed toxicity when sliding a 6mer window along the first 17 nucleotides of all miRtrons (new Figure S7e). This confirms that the nts around positions 2-7 have the highest predicted toxicity when acting as a seed. Looking at non-miRtrons introns as a control is a entire project as the nucleotide composition of introns is very different depending on whether they are first or internal introns, for instance. A detailed analysis of introns would go beyond the scope of this manuscript.

10) miR-449 as backup for miR-34a. If the tumor suppressive function is being selected for, and many different seeds are toxic, why is the exact same seed used as backup? Why are there not more toxic-seed containing tumor suppressor miRNAs? Why are toxic seeds overall rare among endogenous miRNAs and depleted from ancient miRNAs, whereas many tumor suppressing proteins are extremely conserved? Their general toxicity would only be a problem if expression were not restricted to genotoxic stress situations. Perhaps here is another argument for the evolutionary result: old miRNAs tend to have much broader domains of expression...

Response: All good points. There are actually many tumor suppressive miRNAs that seem to act through this mechanism. We just chose the most accepted and major ones to not run into the discussion of pleiotropic effects of miRNAs. Our web site now allows to identify new ones and study them. In addition, as we pointed out there are many different ways for a miRNA to be tumor suppressive. Members of the let-7 and miR-200 families for instance do not carry toxic 6mer seeds and they are tumor suppressive through induction and maintenance of differentiation, general differentiation in case of let-7 and epithelial differentiation in case of miR-200.

In regard to the question for why there could be redundancy at the level of the miRNA seeds. That is an almost philosophical question. After all, there are many miRNAs with the same seed, regardless of their function. We believe that an effective anti-cancer mechanism needs to be redundant by definition to prevent cancer to get around the mechanism by simply mutating the site or by deleting entire gene regions.

11) Fig. S10a: "total read numbers" or "reads per million"? If indeed counts, then error-bars seem small for miR-34b-5p and miR-34c-5p and large for miR-34a-5p. "Variance of two biol. Replicates" should just be replaced by the two data points (swarm plot).

Response: Yes, these are total read counts. The error bars are indeed the variance of the two data points. We have added the actual data points to the plots shown in Fig. S10a and Fig. S10b.

12) The discussion mentions miR-34a loss-of-function experiments with somewhat inconclusive results. Regarding gain-of-function: are cells treated with toxic seed siRNAs more sensitive to genotoxic drugs than cells treated with non-toxic seed siRNAs? Are there synergistic/sensitizing effects? Could this allow to disentangle whether toxic seeds (miR-34a) are downstream of genotoxic triggers or an independent enhancer of cell death? While absolutely not required, this experiment seems doable and could potentially add substantial weight to the oncology aspect of the paper.

Response: We have performed the suggested experiments and found that the toxicity of miR-34a and chemotherapeutic drugs (we tested doxorubicin, etoposide and carboplatin) is not synergistic. This is consistent with chemo tapping into the 6mer seed mechanism. We have added this info to supplementary figure 11.

In addition to the requested changes we have made one more improvement. Upon close inspection of the screening data of the 4096 duplexes in the four cell lines we noticed that the screen of the 3LL cells was a bit off, the cells overall seemed less sensitive particularly to the moderately toxic 6mer seeds. We identified a minor problem in the way the duplexes were handled in this one experiment. As a consequence we have since repeated the entire screen and now the data on the 3LL cells are more consistent with the other three cell lines. As a result we have updated Figs. 1b, 1c (right panel), Fig. 2b (bottom right panels), S2 and S3a as well as the 6merdb.org web site.